# Nucleotide binding halts diffusion of the eukaryotic replicative helicase during activation

Daniel Ramírez Montero [1], Humberto Sánchez [1], Edo van Veen [1], Theo van Laar [1], Belén Solano[1], John F. X. Diffley [2]✉ & Nynke H. Dekker [1]✉

The eukaryotic replicative helicase CMG centrally orchestrates the replisome and leads the way at the front of replication forks. Understanding the motion of CMG on the DNA is therefore key to our understanding of DNA replication. In vivo, CMG is assembled and activated through a cell-cycle-regulated mechanism involving 36 polypeptides that has been reconstituted from purified proteins in ensemble biochemical studies. Conversely, single-molecule studies of CMG motion have thus far relied on pre-formed CMG assembled through an unknown mechanism upon overexpression of individual constituents. Here, we report the activation of CMG fully reconstituted from purified yeast proteins and the quantification of its motion at the single-molecule level. We observe that CMG can move on DNA in two ways: by unidirectional translocation and by diffusion. We demonstrate that CMG preferentially exhibits unidirectional translocation in the presence of ATP, whereas it preferentially exhibits diffusive motion in the absence of ATP. We also demonstrate that nucleotide binding halts diffusive CMG independently of DNA melting. Taken together, our findings support a mechanism by which nucleotide binding allows newly assembled CMG to engage with the DNA within its central channel, halting its diffusion and facilitating the initial DNA melting required to initiate DNA replication.

Eukaryotic DNA replication is catalyzed by a MDa-sized dynamic protein complex known as the replisome. The replisome is powered by the replicative helicase CMG (Cdc45/Mcm2-7/GINS), which centrally orchestrates the other components and leads the way at the front of replication forks[1]. Understanding the motion of CMG on DNA is therefore crucial to our understanding of how cells successfully replicate DNA. In vivo, loading and activation of CMG occur in temporally separated fashion. In *Saccharomyces cerevisiae* in particular, CMG loading occurs at specific sequences known as origins of replication[1]. First, in the G1-phase of the cell cycle, a set of proteins known as 'loading factors' scans the DNA until such origins of replication are located, at which inactive single and double Mcm2-7

hexamers are then loaded onto dsDNA[2–5]. In the subsequent S-phase, double Mcm2–7 hexamers are selectively phosphorylated by the cell cycle-regulated Dbf4-dependent kinase (DDK)[6]. Then, a set of proteins known as 'firing factors' facilitates the assembly of full CMG by recruiting the helicase-activating factors Cdc45 and GINS to the phosphorylated Mcm2–7 double hexamers[7,8]. Upon full assembly, CMG must transition from encircling dsDNA to encircling ssDNA, so that it can unwind dsDNA by steric exclusion of the non-translocation strand[9]. This transition is known as CMG activation and consists of two steps. In the first step, ATP binding allows each CMG in a double hexamer to melt 0.6–0.7 turns of dsDNA within its central channel[10,11]. In the second step, each CMG extrudes one strand of the double helix

[1]Department of Bionanoscience, Kavli Institute of Nanoscience, Delft University of Technology, Delft, The Netherlands. [2]Chromosome Replication Laboratory, Francis Crick Institute, London, UK. ✉e-mail: john.diffley@crick.ac.uk; n.h.dekker@tudelft.nl

from its central channel; this final step requires ATP hydrolysis and the action of the firing factor Mcm10[10]. After having extruded one strand of DNA, each activated sister CMG translocates on ssDNA in a 3′-to-5′ direction by hydrolyzing ATP[1,12,13], allowing the two helicases to bypass and move away from each other[10], and committing the cell to initiate DNA replication[1]. This entire process requires a minimal set of 36 polypeptides and has been fully reconstituted from purified *Saccharomyces cerevisiae* proteins in ensemble biochemical studies[10].

To date, the motion of fully reconstituted CMG has been studied in bulk biochemical assays[10,14] with a temporal resolution of minutes. These ensemble biochemical studies have provided us with important insights into the average behavior of CMG; nonetheless, lower probability behaviors are averaged out in the ensemble readouts. On the other hand, single-molecule studies have the power to isolate and study low probability events with higher temporal resolution[15]; nevertheless, single-molecule studies of CMG motion so far[16–18] have focused on pre-formed CMG assembled through a poorly understood mechanism that requires co-overexpression of individual subunits[19,20]. What is more, given that such pre-formed CMG requires an artificial region of ssDNA to bind DNA[16,19], these studies could not access the intricacies of CMG activation, such as the role of CMG motion during this process. We therefore set out to assemble and activate CMG from purified *Saccharomyces cerevisiae* proteins and study its motion during this process at the single-molecule level.

## Results

### A hybrid ensemble and single-molecule assay to visualize fully reconstituted CMG

One of the biggest challenges of studying the motion of fully reconstituted CMG at the single-molecule level is to prevent all the proteins involved from aggregating onto the long (tens of kbp) DNA molecules needed to observe motion of diffraction-limited fluorescent spots[16,17,21]. To overcome this challenge, we developed a hybrid ensemble and single-molecule assay to (1) assemble and activate fully reconstituted fluorescent CMG onto ~24 kbp DNA molecules containing a natural yeast ARS1 replication origin in an aggregation-free manner; and (2) use a combination of dual optical trapping and confocal scanning microscopy[22] to image and quantify the motion of fluorescent CMG along DNA molecules held in an optical trap (Fig. 1a–i, Supplementary Fig. 1a and "Methods" section). To this end, we added both digoxigenin and desthiobiotin moieties at each end of a linear 23.6 kb DNA containing a natural ARS1 origin of replication. We then bound the functionalized DNA to streptavidin-coated magnetic beads and used it to assemble and activate CMG ("Methods" section). In short, we loaded Mcm2–7 hexamers onto the bead-bound DNA, phosphorylated double Mcm2–7 hexamers with DDK and washed the beads with a buffer solution containing 300 mM KCl. We then assembled *and* activated CMG for 15 min in the presence of fluorescently labeled Cdc45[LD555] (Supplementary Fig. 1a), which supports DNA unwinding near WT levels (Supplementary Fig. 1b). Following CMG assembly and activation, we washed the beads again with a buffer solution containing 300 mM KCl to select for fully mature CMG[7,8], and 'paused' the reaction by removing ATP. DNA:CMG complexes were then eluted from the magnetic beads by competing the desthiobiotin-streptavidin interaction with an excess of free biotin[23]. Following elution, DNA:CMG complexes were tethered between two optically trapped anti-digoxigenin-coated polystyrene beads, and transferred into a buffer solution containing Mcm10, RPA and either ATP, no nucleotide, or the slowly hydrolyzable ATP analog ATPγS. We then scanned the DNA with a confocal scanning laser and observed fluorescent CMG helicases as diffraction-limited spots on the otherwise unlabeled DNA (Fig. 1a–ii). Approximately a third of the trapped DNA molecules contained diffraction-limited fluorescent CMG spots, typically a single one (Fig. 1b). We deduced the number of CMG per diffraction-limited spot by counting the photobleaching steps within each spot

(Supplementary Fig. 3 and "Methods" section). As this showed that most spots contained 1 CMG (Fig. 1c), it followed that most DNA molecules had a total of 1 CMG (Supplementary Fig. 1c), where a priori one might have expected a total closer to 2 or multiples thereof. We consider it unlikely that our experimentally measured lower number is substantially influenced by the labeling efficiency of Cdc45[LD555], which we measured to be 85 ± 4% ("Methods" section); rather, we attribute it to (a potential combination of) other factors including loss of Cdc45 during the high salt washes and downstream handling, CMG dissociation at nicks[24] on the DNA during the ensemble activation, or CMG diffusing off the ends of the DNA during elution. Furthermore, it was recently shown that each Mcm2–7 in a double hexamer independently matures into CMG[8]. Thus, we cannot discard the possibility that in our system only one of the two Mcm2–7 hexamers is fully matured into CMG[25].

### Mature CMG is preferentially assembled near origins of replication

We first looked at the initial positions of CMG on the DNA. Of note, because we cannot differentiate between the two possible orientations of the DNA in our experiments, we display the initial positions of CMG in plots showing the distance from the center of the DNA[2]. We observed a wide distribution of initial positions with a peak near or at the ARS1 origin (Fig. 1d). Furthermore, spots containing two CMG complexes were less widely distributed around the origin than spots containing one CMG (Fig. 1e, f). Taken together, these results are consistent with a preferential assembly of sister CMG helicases near the ARS1 origin, followed by the motion of individual activated helicases away from the origin during the 15-min ensemble activation reaction.

### Colocalization of fluorescent Cdc45 and fluorescent Mcm2–7 hexamers is DDK-dependent

Salt-resistant Cdc45 is considered a hallmark of mature CMG[7,8]. Nonetheless, if the Cdc45[LD555] spots that we observe are part of bona fide CMG, their presence on the DNA should be dependent on DDK[6–8]. To confirm this, we quantified the co-localization of red fluorescently labeled Mcm2–7[JF646-Mcm3] with green fluorescently labeled Cdc45[LD555] (shown to jointly support DNA unwinding (Supplementary Fig. 2a)) in the presence and absence of DDK (Fig. 1g–j). While nearly 20% of Mcm2–7[JF646-Mcm3] spots colocalized with Cdc45[LD555] in the presence of DDK, we observed an ~8-fold decrease in this colocalization in the absence of DDK (Fig. 1h–j). The 20% colocalization that we observe in the presence of DDK is in agreement with previous observations that the in vitro assembly of CMG is less efficient than the loading of Mcm2–7 double hexamers[10,11]. Taken together, these results show that the Cdc45[LD555] fluorescent spots in our images correspond to bona fide CMG. We attribute the residual colocalization of Mcm2–7[JF646-Mcm3] and Cdc45[LD555] in the absence of DDK to non-specific interactions (Supplementary Fig. 2h) and/or to traces of phosphorylated Mcm2–7 in the protein preparation[8].

### Fully reconstituted CMG exhibits two quantitatively distinct motion types

We next sought to quantify the motion of CMG in the presence of ATP. For this, we implemented a change-point algorithm (CPA) to fit linear segments through regions of the position-vs.-time plots of individual spots (Fig. 2a–c and "Methods" section); the slopes of these segments then give us a noise-reduced value of the instantaneous velocities of individual fluorescent spots. To calibrate our analysis, we imaged dCas9[LD555] with the same imaging conditions that we used for CMG (Fig. 2a and Supplementary Fig. 3); because dCas9[LD555] is static on the DNA, it provides us with a measure of the velocity error in our system. After drift correction, the distribution of instantaneous velocities of fluorescent dCas9[LD555] spots after the CPA fit is centered at 0 bp/s and

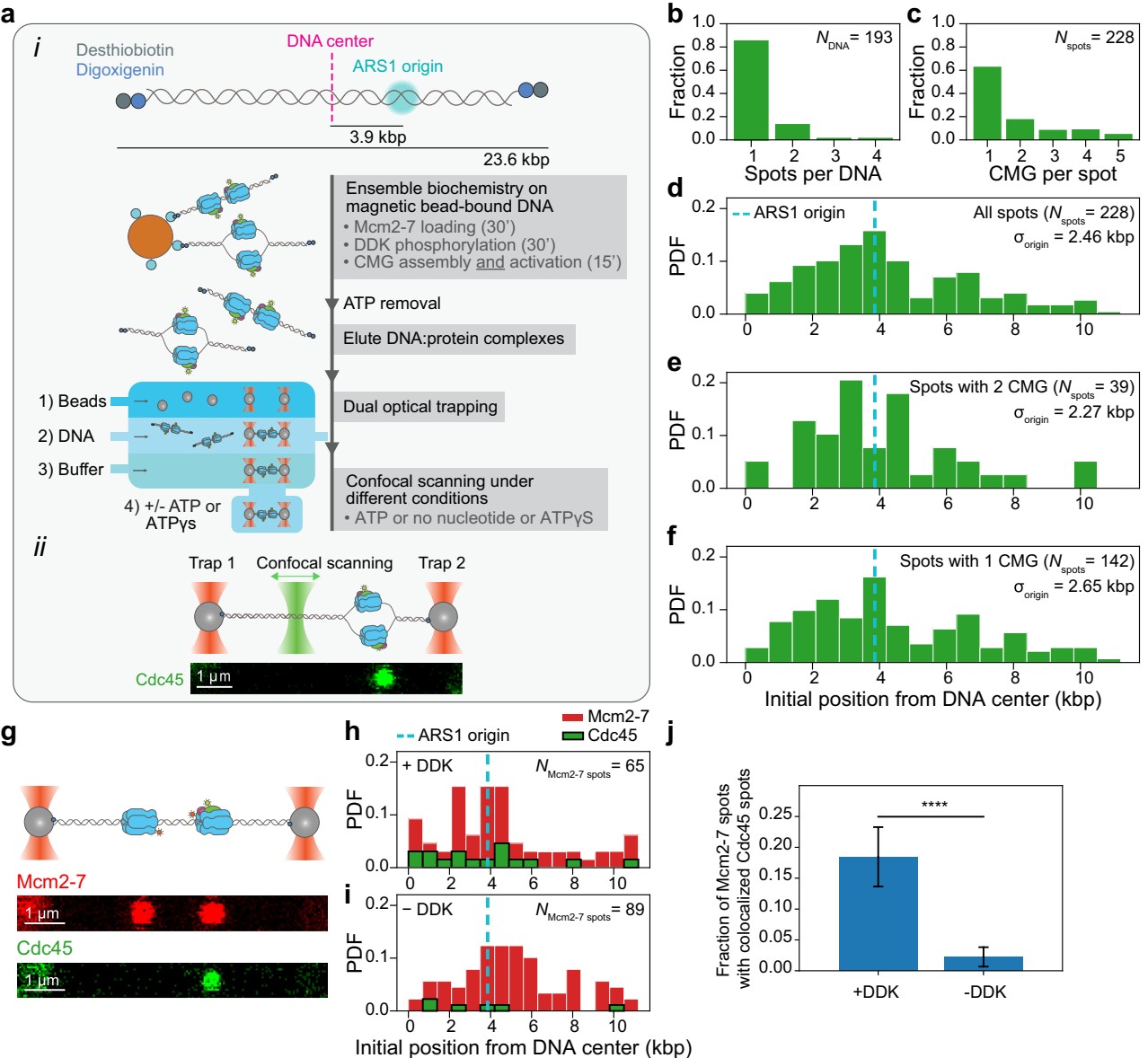

**Fig. 1 | Single-molecule imaging of fully reconstituted CMG. a–i** Description of hybrid ensemble and single-molecule assay to image fully reconstituted CMG. **a-ii** Example scan of an optically trapped DNA molecule containing one CMG diffraction-limited spot. **b** Distribution of numbers of CMG diffraction-limited spots per DNA. **c** Distribution of numbers of CMG complexes within each diffraction-limited spot. **d–f** Distribution of initial positions on the DNA of **d** all CMG diffraction-limited spots, **e** diffraction-limited spots containing 2 CMG complexes or **f** diffraction-limited spots containing 1 CMG complex; the ARS1 origin of replication is indicated by the dashed cyan line. **g** Example scans separately showing Mcm2-7[JF646] diffraction-limited spots (top) and Cdc45[LD555] diffraction-limited spots (bottom) on the same DNA molecule. **h, i** Distributions of initial positions of Mcm2-7[JF646] spots and Cdc45[LD555] spots on the DNA in the **h** presence or **i** absence of DDK. In each condition (with or without DDK), the histograms of Mcm2-7[JF646] and Cdc45[LD555] initial positions are weighted by the total number of Mcm2-7[JF646] spots. **j** Mean fraction of Mcm2-7[JF646] diffraction-limited spots with colocalized Cdc45[LD555] diffraction-limited spots in the presence ($N_{Mcm2-7\ spots}$ = 65) or absence ($N_{Mcm2-7\ spots}$ = 89) of DDK; error bars show the standard error of proportion. Statistical significance was obtained from a two-sided binomial test (p-value = $2.2 \times 10^{-8}$).

has a width $\sigma_{dCas9}$ that reflects our experimental uncertainty in velocity measurement (Fig. 2a inset). For all CMG motion analysis, we defined a conservative velocity cutoff of $5 \times \sigma_{dCas9}$ (=2.0 bp/s) to categorize fluorescent spots as static or mobile; we considered mobile any fluorescent spot with at least one CPA segment with a slope above this threshold, and all other spots static.

Following the approach described above, we determined that ~70% of CMG spots are mobile when imaged in a buffer solution containing RPA, Mcm10 and ATP (Fig. 2b, d and Supplementary Fig. 5a). Unexpectedly, when we imaged CMG in a buffer solution containing RPA, Mcm10 and no ATP, we observed that ~40% of CMG spots were also mobile (Fig. 2c, d and Supplementary Fig. 5d). Nonetheless, we noticed qualitative differences in the motion of CMG in the presence

and absence of ATP: while CMG seemed to move unidirectionally in the presence of ATP (Fig. 2b and Supplementary Movie 1), it appeared to move in a more random (e.g. diffusive) manner in the absence of ATP (Fig. 2c and Supplementary Movie 2). To quantitatively characterize these two apparently distinct motion types, we employed two independent approaches. First, we looked at the CPA segments of all the traces in each condition and calculated the probability that consecutive segments have the same direction (Fig. 2e); in the absence of noise, this probability should equal 1 for unidirectional motion, and 0.5 for random motion. As seen in Fig. 2e, our measured probabilities closely match these expected values, providing quantitative underpinning of our initial observations. As an independent approach, we conducted anomalous diffusion analysis of the mobile traces in each

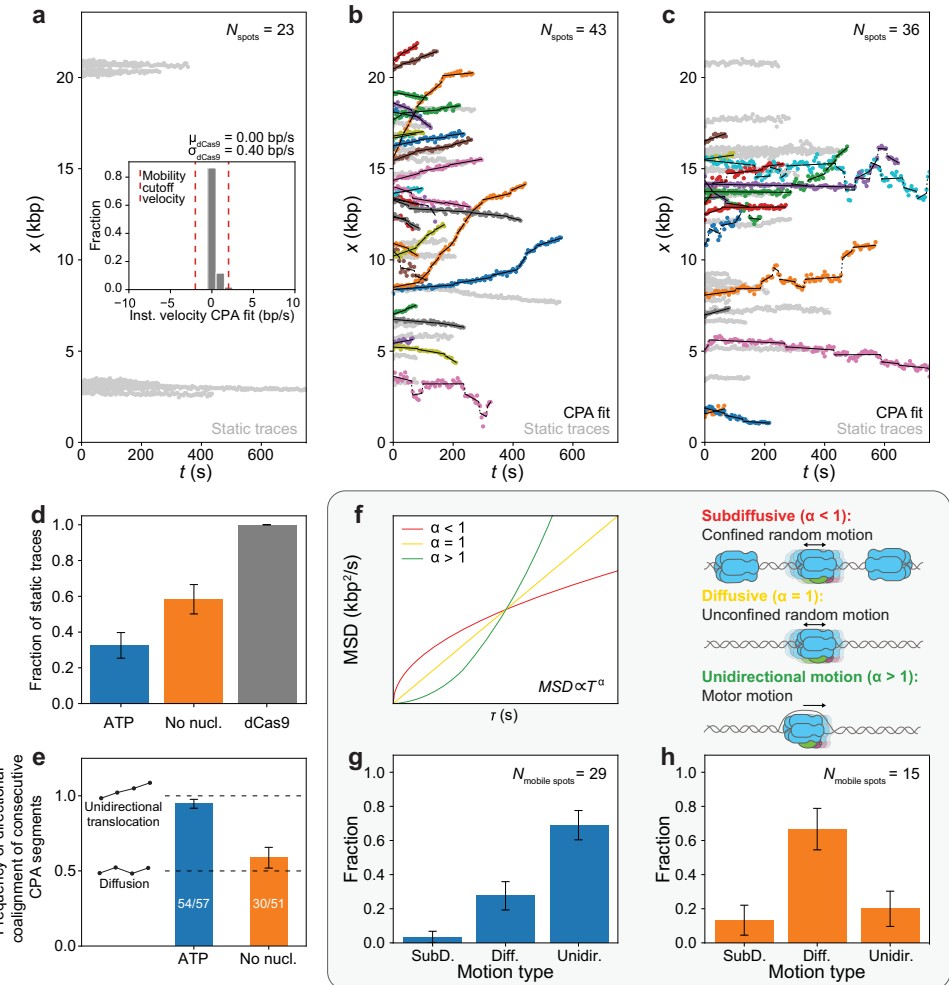

**Fig. 2 | Fully reconstituted CMG exhibits two different motion types. a** Position vs. time plots of dCas9[LD555] spots; (inset) distribution of instantaneous velocities coming from the CPA fits of dCas9[LD555] spots; red lines show the instantaneous velocity cutoff ($5\sigma_{dCas9}$) used to separate CMG spots in **b** and **c** into static or mobile; CPA fits are not shown for clarity. **b**, **c** Position vs. time plots of CMG spots in the **b** presence of ATP or **c** absence of nucleotide; CPA fits are plotted in black, static traces are shown in light gray. **d** Ratio of static CMG traces in the presence of ATP ($N_{spots} = 43$), absence of nucleotide ($N_{spots} = 36$), and static dCas9 ($N_{spots} = 23$) traces; error bars show the standard error of proportion. **e** Frequency of consecutive CPA segments with the same direction for CMG spots in the presence of ATP

($N_{mobile\ spots} = 29$) or absence of nucleotide ($N_{mobile\ spots} = 15$); inset diagrams illustrate expected segment directions of a unidirectionally moving spot (top) or a diffusive spot (bottom); error bars show the standard error of proportion. **f** (left panel) Idealized examples of MSD *vs.* delay time τ plots with an anomalous coefficients $\alpha < 1$ (red), $\alpha = 1$ (yellow), and $\alpha > 1$ (green); (right panel) diagrams illustrating the types of CMG motion corresponding to each of these three cases: constrained diffusion ($\alpha \ll 1$), free diffusion ($\alpha \approx 1$) or unidirectional motion ($\alpha \gg 1$). **g**, **h** Fraction of mobile CMG traces classified into different motion types in the **g** presence of ATP or **h** absence of nucleotide; error bars show the standard error of proportion.

condition (Fig. 2f–h and Supplementary Fig. 5b, e). For each individual trace, we calculated the mean-squared displacement (MSD) as a function of the lag time τ, and then fitted the result to the equation $MSD(\tau) \propto \tau^\alpha$ to extract the anomalous diffusion coefficient $\alpha$ ("Methods" section). The value of $\alpha$ then allowed us to classify each trace into different motion types, as $\alpha \gg 1$ for unidirectionally moving molecules, $\alpha \approx 1$ for freely diffusive molecules, and $0 < \alpha \ll 1$ for molecules undergoing constrained diffusion (Fig. 2f). This anomalous diffusion analysis confirmed that unidirectional motion is most likely when ATP is present (Fig. 2g), whereas diffusive behavior is most likely when ATP is absent (Fig. 2h).

We note that we observed a small population of seemingly diffusive CMG spots in the presence of ATP, and a small population of seemingly unidirectionally moving CMG spots in the absence of ATP (Fig. 2g, h). We hypothesized that these subpopulations might have arisen from misclassification of short traces[26]. To test this hypothesis, we simulated two populations of single-molecule traces of varying lengths within the range of our experimental data: one population

solely consisting of unidirectionally moving traces, and the other population solely consisting of freely diffusive traces ("Methods" section). We then carried out the same anomalous diffusion analysis that we did on the experimental CMG data on both simulated data sets. We observed that the distribution of motion types for the simulated unidirectional traces looked very similar to that of the experimental mobile CMG traces in the presence of ATP (Supplementary Fig. 6a and Fig. 2g), whereas the distribution of motion types for the simulated diffusive traces looked very similar to that of the experimental mobile CMG traces in the absence of ATP (Supplementary Fig. 6b and Fig. 2h). Thus, the results of our simulations suggest that, overall, the mobile traces in the presence of ATP represent unidirectional motion, whereas the mobile traces in the absence of ATP represent diffusive motion.

## Analysis of CMG motor motion
Following this identification of two distinct types of CMG mobility, we investigated both in further depth. We first investigated the

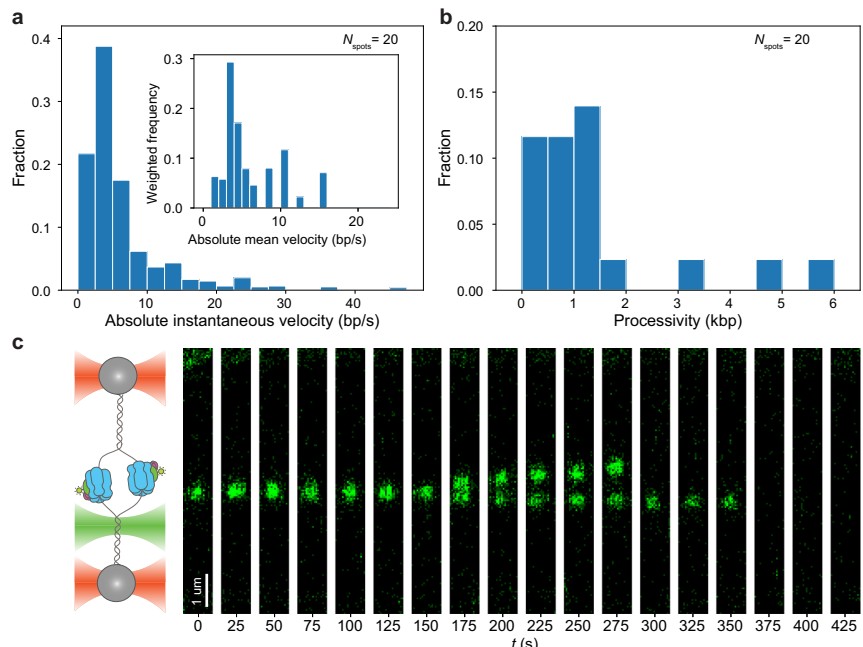

**Fig. 3 | Analysis of CMG motor motion. a** Distribution of absolute instantaneous velocities of unidirectionally moving CMG spots in the presence of ATP; (inset) Distribution of absolute mean velocities of unidirectionally moving CMG spots in the presence of ATP normalized by the length of each trace. **b** Distribution of processivities of unidirectionally moving CMG spot in the presence of ATP. **c** Example kymograph of two unidirectionally moving CMG spots that start within the same diffraction-limited spot and split up into two distinct diffraction-limited spots that move along the DNA in opposite directions ($N_{splitting\ events} = 2$).

unidirectional motor motion of CMG, the motion type that powers the replisome. We thus specifically analyzed the velocities of unidirectionally moving CMG spots in the presence of ATP, which yielded a distribution of instantaneous velocities with a peak at ~5 bp/s (Fig. 3a), consistent with previous single-molecule studies on preformed CMG in the presence of RPA[17]. This distribution has a long tail, reaching up to instantaneous velocities of ~45 bp/s. These higher velocities are low in probability, suggesting that CMG can only achieve these high velocities in short time bursts. Consistent with this, when we calculated the time-averaged velocity of each CMG spot and examined the resulting distribution (Fig. 3a inset), we did not observe such high velocities. The distribution of time-averaged velocities has a peak at ~5 bp/s, which is consistent with ensemble biochemical studies of CMG motion[14]. Notably, we did not observe any noticeable backtracking of CMG (Fig. 2b), which is consistent with previous studies suggesting that RPA prevents CMG backtracking by keeping the lagging strand template out of the central channel of CMG[17].

When analyzing the unidirectional traces in the presence of ATP, we also observed a few instances of two unidirectionally translocating CMGs initially located within the same diffraction-limited spot, but that then split from one another and give rise to two diffraction-limited spots of half the intensity of the original spot (Fig. 3c and Supplementary Movie 3). These observations are consistent with in vitro biochemical studies of helicase activation[10] showing that sister CMGs move in opposite directions upon their activation. The low probability of these splitting events is to be expected because (i) we allow CMG to become activated and translocate on the DNA for 15 min before imaging, and (ii) because most DNA molecules contain 1 CMG (Supplementary Fig. 1c).

## Nucleotide binding halts diffusive CMG

Our data shows that CMG diffuses on DNA in the absence but not in the presence of ATP (Fig. 2b, c, g, h), suggesting that ATP is involved in stopping the diffusive motion of CMG. To investigate whether it was the binding or the hydrolysis of ATP that stopped the diffusive motion

of CMG, we investigated CMG motion in a buffer solution supplemented with RPA, Mcm10 and the slowly hydrolysable ATP analog ATPγS. When we imaged CMG under these conditions, the vast majority of CMG spots were found to be static (Fig. 4a, b and Supplementary Fig. 5i). Taken together with our data in the presence and absence of ATP (Fig. 2b–d), our results show that it is the nucleotide binding and not the hydrolysis that halts the diffusive motion of CMG. Furthermore, our results confirm that ATP hydrolysis is required for the unidirectional translocation of CMG[13].

Previous biochemical studies showed that ATP binding allows newly formed CMG to melt 0.6–0.7 turns of the DNA within its central channel[10], which was recently confirmed by cryo electron microscopy[11]. Comparing these previous observations with our single-molecule results led us to hypothesize that 1) the diffusive motion that we observed in the absence of ATP corresponded to CMG surrounding dsDNA, and that 2) the halting of such diffusive motion in the presence of ATPγS is due to CMG melting the DNA within its central channel.

To test whether CMG can diffuse on dsDNA in the absence of ATP, we developed an ensemble CMG sliding assay (Fig. 4d, e and "Methods" section). Briefly, we synthesized two 1.4 kb linear DNA constructs biotinylated at one end and containing an ARS1 origin. The non-biotinylated end of the constructs was then either left as a free end or covalently crosslinked to a M.HpaII methyltransferase[27]. Because the crosslinked methyltransferase is too large to fit inside the central channel of CMG[28,29], it should stop CMG from diffusing off the end of the DNA, which would otherwise be free to diffuse off the free end. We then bound both DNA constructs to streptavidin-coated magnetic beads, and assembled CMG in the presence of fluorescent Cdc45[LD555] onto them; importantly, we omitted the firing factor Mcm10 from the activation reaction to prevent strand extrusion from the central channel of CMG and ensure that CMG is surrounding dsDNA[10]. After CMG assembly, we incubated the bead-bound DNA in a buffer solution with or without ATPγS, and monitored the amount of fluorescent Cdc45[LD555] present on the DNA over time. As seen in Fig. 4d, e, we detected the fastest decay of Cdc45[LD555] signal in the DNA construct

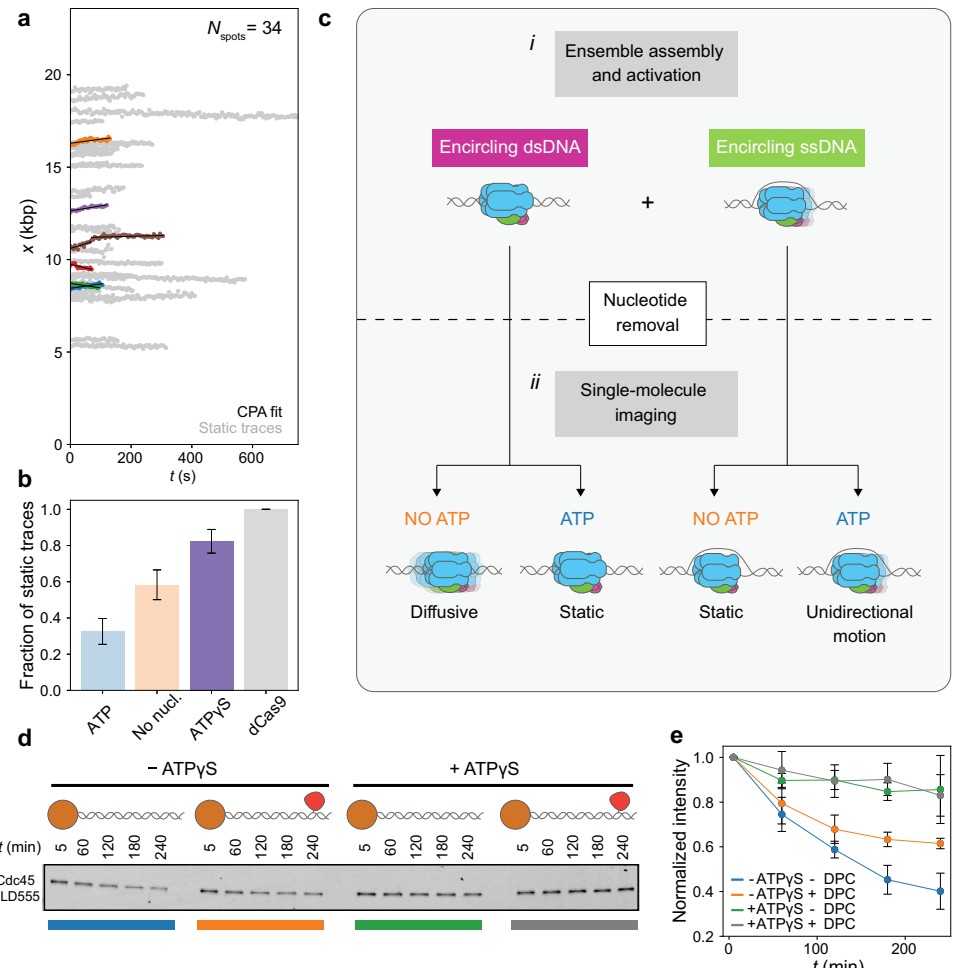

**Fig. 4 | Nucleotide binding halts CMG diffusion. a** Position vs. time plots of CMG spots in the presence of ATPγS; CPA fits are plotted in black, static traces are shown in light gray and mobile traces are shown in all other colors. **b** Fraction of static CMG spots in the presence of ATPγS ($N_{spots}$ = 34); the results from Fig. 2d, are shown as light bars for comparison; error bars show the standard error of proportion. **c** Model proposed to explain experimental motion results; **c-i** proposed two populations of CMG present in the ensemble CMG activation reaction; **c-ii** summary of experimental outcomes in Figs. 2 and 4a, and proposed explanation of their origins. **d** Fluorescent scan of an SDS-PAGE gel showing the amount of Cdc45LD555 left on linear DNA bound to magnetic beads at one end and containing either a free end or an end capped with a covalently crosslinked methyltransferase. **e** Densitometry quantification of the experiment shown in **d** showing the average normalized intensity of three replicates together with their standard deviation. Data points are connected by solid lines to guide the eye. Source data are provided as a Source Data file.

with a free end in the absence of ATPγS, whereas there was only a small decay in the same DNA construct when ATPγS was present. We also observed some decay, albeit smaller in magnitude, in the DNA construct with a capped end in the absence of ATPγS (Fig. 4d, e); this smaller decay may result from spontaneous opening of the Mcm2–7 ring of CMG in the absence of nucleotide[16,30] on such timescales. Thus, the results of our ensemble assay support our hypothesis that CMG diffuses on dsDNA in the absence of ATP, and that nucleotide binding halts this diffusive motion.

To test whether the nucleotide-binding-mediated halting of CMG diffusion was due to DNA melting within the central channel of CMG, we employed a recently reported Mcm2 mutant in which six residues directly involved in DNA melting are substituted with alanine (hereafter referred to as Mcm2[6A])[11]. Notably, Mcm2[6A] supports CMG assembly but does not support either DNA melting or strand extrusion[11], allowing us to separate the effect of DNA melting from DNA binding. Thus, if DNA melting by CMG is indeed what halts its diffusive motion, then CMG assembled with Mcm2[6A] should be fully diffusive in the presence of ATP. Nevertheless, the vast majority of CMG spots assembled with Mcm2[6A] are static in the presence of ATP (Supplementary Fig. 9a, f). Furthermore, when we performed our

ensemble CMG sliding assay with Mcm2[6A], we observed the same trends as with WT Mcm2 (Supplementary Fig. 9g, h), providing further evidence that CMG diffuses on dsDNA and that DNA melting is not necessary for the halting of CMG diffusion after ATP-binding. Altogether, our single-molecule and ensemble biochemical data show that the halting of CMG diffusion in the presence of nucleotide is not due to DNA melting by CMG, but due to binding of CMG to the DNA via other Mcm2-7:DNA interactions[11].

## Discussion

We report the single-molecule motion quantification of fully reconstituted CMG assembled at an origin of replication. To enable this, we developed a hybrid ensemble and single-molecule assay based on the double functionalization of DNA ends, which we validated by verifying that the number of fluorescent CMG helicases per diffraction-limited spot, the initial CMG positions on the DNA, and the DDK-dependent colocalization of fluorescently labeled Cdc45 and Mcm2-7 agree with the consensus established by previous biochemical observations (Fig. 1).

In our motion analysis, we observe a static and a mobile population of CMG both in the presence and in the absence of ATP (Fig. 2 and

Supplementary Fig. 5). Nonetheless, we quantitatively show that the mobile population in the presence of ATP preferentially moves unidirectionally, as expected for a molecular motor, whereas the mobile population in the absence of ATP preferentially moves in a diffusive manner (Fig. 2). To explain these motion outcomes, we propose the model summarized in Fig. 4c, which is based on the assumption that our ensemble activation reaction contains a mixture of two populations of CMG: one population encircling dsDNA, and another population encircling ssDNA (Fig. 4c–i). In the first part of our model, we propose that the population of CMG encircling ssDNA moves unidirectionally in the presence of ATP (as the motor can then hydrolyze ATP to unwind DNA), but remains static in the absence of ATP (as the motor is unable to unwind DNA without ATP hydrolysis[9,10,16]) (Fig. 4c, right half). This part of the model is supported by the fact that we observe very similar instantaneous and average velocities of unidirectionally translocating CMG spots to those of previous single-molecule and ensemble biochemical studies of DNA unwinding[14,17,18]. We consider an alternative scenario in which at least part of the unidirectional translocation that we observe in the presence of ATP corresponds to CMG translocating on dsDNA (Supplementary Fig. 10) – as was postulated to occur for CMG helicases that bypassed each other after the collision of two replication forks or after encountering a flush ss/dsDNA junction[31–33] – to be less likely. Consider, for example, the CMG splitting events that we observe (Fig. 3c). Because CMG translocation on dsDNA has the same 3′-to-5′ polarity as on ssDNA[33], one could postulate that the splitting events represent two helicases that surround dsDNA following assembly at a single Mcm2-7 double hexamer, but such helicases would move *towards* each other (remaining within a single diffraction-limited spot), and not *away* from each other as our data shows. One could also postulate that the splitting events represent two helicases encircling dsDNA following assembly at distinct Mcm2-7 double hexamers within a single diffraction-limited spot. However, our analysis of Mcm2-7 spots shows such occurrences to be unlikely (Supplementary Fig. 2c). Further studies will be required to assess whether and how the unidirectional motion of CMG differs according to whether it occurs on ssDNA or dsDNA.

In the second part of our model, we propose that the population of CMG encircling dsDNA loses its bound ATP when we remove ATP from the buffer following our ensemble activation (Fig. 1a); ATP dissociation then causes CMG to disengage from the DNA inside its central channel and thereby become diffusive. This diffusive population remains so in the absence of nucleotide (giving rise to the diffusive population we observe in the absence of nucleotide); nevertheless, upon ATP re-addition and re-binding, CMG is allowed to re-engage with the DNA inside its central channel and thus become static (giving rise to the static population in the presence of ATP). In support of this hypothesis, we showed that CMG can diffuse on dsDNA and that nucleotide binding halts this diffusion (Fig. 4a, b). Furthermore, we show that this halting occurs independent of the previously observed DNA melting by CMG upon nucleotide binding[10,11] (Supplementary Fig. 9), suggesting that DNA engagement and DNA melting by CMG need not occur concomitantly. We propose that the presence of ATP in the nucleus prevents newly assembled CMG from diffusing along the DNA, poising the helicase to catalyze the initial melting required to initiate replication. Further studies investigating which of the additional Mcm2-7:DNA contacts within CMG[11] are responsible for halting CMG diffusion upon nucleotide binding will shed further light into our observations.

Our measured diffusion coefficient of freely diffusive CMG under the ionic strength conditions of this study (250 mM K-glutamate) (Supplementary Fig. 5l) is similar to our previously measured diffusion coefficient of single Mcm2-7 hexamers in higher ionic strength conditions (500 mM NaCl)[2]. This observation suggests that CMG diffuses more freely on the DNA than single Mcm2-7 hexamers, which in turn suggests that CMG has fewer contacts with the DNA than Mcm2-7

hexamers as recently confirmed by structural studies[11,34]. We also note that single-molecule studies with pre-formed *D. melanogaster* CMG showed no evidence of CMG diffusion in the presence of ATP[17,18], in agreement with our observations. However, single-molecule studies with pre-formed *S. cerevisiae* CMG reported extensive diffusive behavior in the presence of ATP[16]. Further studies will be required to investigate the reasons for this discrepancy, and to probe potential differences between fully reconstituted and pre-formed CMG, such as whether the phosphorylation state of Mcm2-7 is different, and whether it could influence the interactions that Mcm2-7, the substrate of DDK[1,6], have with the DNA.

ATP-dependent switching of motion modes, as we observe here for CMG, has been previously observed to act differently for other protein complexes, such as Type III restriction enzymes[35,36] and the yeast chromatin remodeler SWR1[37]. In the case of Type III restriction enzymes, ATP hydrolysis triggers diffusion on the DNA[35,36], whereas in the case of SWR1, ATP binding triggers its diffusion along the DNA[37]. Nonetheless, these behaviors are different than in the case of CMG, whose diffusion is halted and not promoted by nucleotide binding.

Given the complexity and the number of components required to fully reconstitute CMG assembly and activation, in vitro single-molecule studies quantifying CMG motion have so far relied on preformed CMG assembled through an unknown mechanism upon overexpression of individual constituents[19,20]. Although these studies have provided us with very important insights into how CMG works, it is unknown whether the assembly mechanism of pre-formed CMG is cell-cycle regulated. It is therefore also unknown whether pre-formed CMG has a similar phosphorylation state to the one it has in vivo. In addition, pre-formed CMG requires an artificial region of ssDNA to bind DNA and thus does not allow us to access the intricacies of CMG assembly and activation. In this study, we quantify the motion of fully reconstituted CMG at the single-molecule level. We believe that the full reconstitution of CMG makes it more likely that the phosphorylation state of its constituents more closely mimics what happens in vivo; this in turn makes it more likely that fully reconstituted CMG can support all relevant interactions with proteins intrinsic and accessory to the replisome. Furthermore, the reliable assay we developed will allow us and others to address important questions regarding CMG motion, such as the mechanistic role of elongation factors, with unprecedented resolution. Finally, we anticipate that the hybrid assay we developed, based on the double functionalization of DNA ends with two orthogonal attachment types, will facilitate the study at the single-molecule level of similarly complex biochemical reactions involved in other nucleic acid:protein interactions.

## Methods
### Biological materials and preparation
#### Molecular cloning and strain generation
**Cdc45-iS6-i2XFLAG.** To generate fluorescently labeled Cdc45, we introduced an 'S6' peptide[38] with an additional glycine linker (GDSLSWLLRLLNG) after residue E197 and immediately before the internal 2XFLAG tag of Cdc45. This was achieved by modifying plasmid pJY13 with a Q5® Site-Directed Mutagenesis Kit (NEB # E0554S) and mutagenic primers DRM_005 and DRM_006. The resulting plasmid pDRM19-01 was transformed into chemically competent One Shot™ TOP10 *E. coli* (Invitrogen # C404010). The sequence of Cdc45-iS6-i2XFLAG was verified by sequencing the entire ORF. *S. cerevisiae* strain yDRM2 overexpressing Cdc45-iS6-i2XFLAG was then generated by linearizing plasmid pDRM19-01 with *Nhe*I-HF® (NEB # R3131S) and transforming it into yeast strain yJF1 (*MAT***a**, W303 background).

**6XHis-S6-dCas9-Halo.** To generate fluorescently labeled dCas9, we introduced an 'S6' peptide with an additional GSS linker (GDSLSWLLRLLNGGS) next to the N-terminal 6XHis tag. This was achieved by modifying plasmid pET302-6His-dCas9-halo with Q5® Site-

Directed Mutagenesis Kit (NEB # E0554S) and mutagenic primers DRM_184 and DRM_185. The resulting plasmid, pDRM21-01 was transformed into NEB® 5-alpha Competent *E. coli* (High Efficiency) cells (NEB # C2987I). The sequence of pDRM21-01 was then verified by sequencing the entire ORF. Plasmid pDRM21_01 was then transformed into *E. coli* BL21-Codon Plus (DE3)-RIL Competent Cells (Agilent # 230245).

**pGL5O-ARS1**. The ARS1 sequence was amplified by PCR using plasmid p5.8kb-ARS1[7] as a template together with primers TL-033 and TL-034, which contained *Asc*I restriction sites at both ends. This PCR product was then cloned into the dephosphorylated *Mlu*I site of the 21.2 kb plasmid pSupercos1-lambda 1,2[39]. The whole sequence of the ARS1 insert was confirmed by sequencing.

**Protein purification.** Cdc6, Mcm10, dCas9-Halo, S6-dCas9-Halo, and Sfp phosphopantetheinyl transferase were expressed in *E. coli* BL21-Codon Plus (DE3)-RIL Cells (Agilent # 230245). GINS was expressed in *E. coli* Bl21 (DE3) Rosetta pLysS Cells (Novagen). Unless otherwise specified, cells were grown to OD600 = 0.40–0.60, induced with 400 μM IPTG for 16 h at 17 °C, and harvested by centrifugation. Pellets were resuspended in 40 mL of lysis buffer and sonicated on ice in a Qsonica Q500 sonicator for 2 min in 5 s on / 5 s off cycles and an amplitude of 40%. Following sonification, the lysate was centrifuged at 8820 g for 20 min in a Beckman-Coulter Avanti JXN 26 centrifuge with rotor JA17, and the clarified supernatant was used for affinity pulldowns.

ORC, Mcm2-7/Cdt1, Polε, Dpb11, Sld2, Sld3/7, Cdc45, Cdc45-iS6, RPA, DDK, and S-CDK were expressed in *S. cerevisiae* strain yJF1 *MAT*a Δpep1 Δbar1 from a bi-directional galactose-inducible promotor. Prior to expression, cells were inoculated at a density of $2 \times 10^5$ cells/ml in YP medium supplemented with 2% raffinose (Carbosynth # OR06197) and 50 mg/ml ampicillin (Merck-Sigma # A9518), and grown overnight at 30 °C to a density of $3-5 \times 10^7$ cells/ml. For S-CDK, 5 μg/ml Nocodazole (Merck-Sigma # M1404), and 2% galactose (Carbosynth # MG05201) were added to the medium and the cells were induced for 3 h before harvesting. For DDK, 2% galactose (Carbosynth # MG05201) was added to the medium and the cells were induced for 3 h before harvesting. For all other proteins, cells were arrested in G1 phase for 3 h before induction with 100 ng/ml of α mating factor (Tebu-bio # 089AS-60221); cells were then induced with 2% galactose (Carbosynth # MG05201) for 3 h before harvesting.

Cells were harvested by centrifugation and washed with lysis buffer. After centrifugation, cells were suspended in lysis buffer supplemented with protease inhibitors (cOmpleteTM EDTA-free Protease Inhibitors (Merck-Sigma #5056489001) and 0.3 mM phenylmethylsulfonyl fluoride (PMSF)), and dropwise dropped into liquid nitrogen. The frozen droplets were grounded in a 6875 SPEX freezer mill for six cycles (run time 2 min and cool time 1 min, with a rate of 15 cps). Following purification, all protein concentrations were determined with Bio-Rad Protein Assay Dye Reagent (Bio-Rad #5000006).

Purifications of ORC, Cdc6, Mcm2-7/Cdt1, Mcm2-7^Halo-Mcm3/Cdt1, Mcm2-7^Mcm2(6A)/Cdt1, and dCas9-Halo have been described previously[2,11].

**DDK**. DDK with a CBP-TEV tag on Dbf4 was purified from *S. cerevisiae* strain ySDK8[40]. Powder was suspended in DDK lysis buffer (25 mM HEPES-KOH pH 7.6. 0.05% NP40 substitute, 10% glycerol, 400 mM NaCl, and 1 mM DTT) supplemented with protease inhibitors. Lysate was cleared in a Beckman-Coulter ultracentrifuge (type Optima L90K with rotorTI45) for 1 h at 235,000 × *g* and 4 °C. The cleared lysate was supplemented with CaCl2 to a final concentration of 2 mM and incubated for 1 h at 4 °C with washed Sepharose 4B Calmodulin beads (GE Healthcare # 17-0529-01) in a spinning rotor. The beads were washed 10 times with 5 ml of DDK-binding buffer (25 mM HEPES-KOH pH 7.6, 0.05% NP40 substitute, 10% glycerol, 400 mM NaCl, 2 mM CaCl2, and

1 mM DTT), and protein was eluted from the beads with DDK elution buffer (25 mM HEPES-KOH pH 7.6, 0.05% NP40 substitute, 10% glycerol, 400 mM NaCl, 2 mM EDTA, 2 mM EGTA, and 1 mM DTT). DDK-containing fractions were pooled and dephosphorylated with 20,000 units lambda phosphatase (NEB # P0753S) for 16 h at 4 °C. The sample was then concentrated in an Amicon Ultra-4 Ultracell 30 kDa centrifugal filter (Merck-Millipore # UFC803024), and injected into a Superdex 200 Increase 10/300 GL column (Cytiva # 15182085) equilibrated in DDK GF buffer (25 mM HEPES-KOH pH 7.6, 0.02% NP40 substitute, 10% glycerol, and 200 mM K glutamate). Positive fractions were pooled and concentrated in an Amicon Ultra-4 Ultracell 30 kDa centrifugal filter (Merck-Millipore # UFC803024). Aliquots were snap frozen and stored at −80 °C.

**GINS**. GINS with a N-terminal 6xHis-tag on Psf3 was purified from *E. coli Rosetta*. Cleared lysate was incubated with Ni-NTA agarose (Qiagen # 30210) for 1 h at 4 °C in GINS lysis buffer (25 mM Tris-HCl pH 7.2, 0.02% NP40 substitute, 10% glycerol, 400 mM NaCl, 10 mM imidazole, and 1 mM DTT) supplemented with protease inhibitors. The beads were washed 5 times with 5 ml of GINS lysis buffer and 5 times with 5 ml of GINS wash buffer (25 mM Tris-HCl pH 7.2, 0.02% NP40 substitute, 10% glycerol, 100 mM NaCl, 15 mM imidazole, and 1 mM DTT). Protein was then eluted from the beads with GINS elution buffer (25 mM Tris-HCl pH 7.2, 0.02% NP40 substitute, 10% glycerol, 100 mM NaCl, 200 mM imidazole, and 1 mM DTT). GINS-containing fractions were pooled and dialyzed against GINS dialysis buffer I (25 mM Tris-HCl pH 7.2, 0.02% NP40 substitute, 10% glycerol, 100 mM NaCl, and 1 mM DTT). The sample was then flowed through a 0.2 micron filter and injected into a MonoQ 5/50 GL column (GE Healthcare # 17-5166-01) equilibrated in GINS MonoQ buffer A (25 mM Tris-HCl pH 7.2, 0.02% NP40 substitute, 10% glycerol, 100 mM NaCl, and 1 mM DTT). GINS was eluted from the column in a 30 CV NaCl gradient from 100 mM to 500 mM. Positive fractions were pooled, concentrated in an Amicon Ultra-4 Ultracell 30 kDa centrifugal filter (Merck-Millipore # UFC803024), and injected into a Superdex 200 Increase 10/300 GL column (Cytiva # 15182085) equilibrated in GINS GF buffer (25 mM Tris-HCl pH 7.2, 0.02% NP40 substitute, 10% glycerol, 150 mM NaCl, and 1 mM DTT). Positive fractions were pooled and concentrated in an Amicon Ultra-4 Ultracell 30 kDa centrifugal filter (Merck-Millipore # UFC803024). Aliquots were snap frozen and stored at −80 °C.

**Polε**. Polε with a C-terminal CBP-TEV tag on Dpb4 was purified from *S. cerevisiae* strain yAJ2[7]. Powder was suspended in Polε lysis buffer (25 mM HEPES-KOH pH 7.6, 10% glycerol, 400 mM KOAc, and 1 mM DTT) supplemented with protease inhibitors. Lysate was cleared in a Beckman-Coulter ultracentrifuge (type Optima L90K with rotor TI45) for 1 h at 235,000 × *g* and 4 °C. The cleared lysate was supplemented with CaCl2 to a final concentration of 2 mM and incubated for 1 h at 4°C with washed Sepharose 4B Calmodulin beads (GE Healthcare # 17-0529-01) in a spinning rotor. The beads were washed 10 times with 5 ml of Polε binding buffer (25 mM HEPES-KOH pH 7.6, 10% glycerol, 400 mM KOAc, 2 mM CaCl2, and 1 mM DTT) and protein was eluted from the beads with Polε elution buffer (25 mM HEPES-KOH pH 7.6, 10% glycerol, 400 mM KOAc, 2 mM EDTA, 2 mM EGTA, and 1 mM DTT). Positive fractions were pooled, passed through a 0.2 micron filter and injected into a 1-ml heparin column (GE Healthcare # 17-0406-01) equilibrated in Polε heparin buffer A1 (25 mM HEPES-KOH pH 7.6, 10% glycerol, 400 mM KOAc, and 1 mM DTT). The column was washed with Polε heparin buffer A2 (25 mM HEPES-KOH pH 7.6, 10% glycerol, 450 mM KOAc, and 1 mM DTT) and Polε was eluted from the column in a 30 CV KOAc gradient from 450 mM to 1000 mM. Peak fractions were pooled, concentrated in an Amicon Ultra-4 Ultracell 30 kDa centrifugal filter (Merck-Millipore # UFC803024), and injected into a Superdex 200 Increase 10/300 GL column (Cytiva # 15182085) equilibrated in Pol

ε GFbuffer(25 mM HEPES-KOH pH 7.6, 10% glycerol, 500 mM KOAc, and 1 mM DTT). Peak fractions were pooled and concentrated in an Amicon Ultra-4 Ultracell 30 kDa centrifugal filter (Merck-Millipore # UFC803024). Aliquots were snap frozen and stored at −80 °C.

**S-CDK.** S-CDK with an N-terminal CBP-TEV tag on Clb5 Δ1-100 was purified from *S. cerevisiae* (Lucy Drury, Francis Crick Institute, unpublished). Powder was suspended in S-CDK lysis buffer (40 mM HEPES-KOH pH 7.6, 0.01% NP40 substitute, 10% glycerol, and 300 mM KOAc) supplemented with protease inhibitors. The lysate was cleared in a Beckman-Coulter ultracentrifuge (type Optima L90K with rotorTI45) for 1 h at 235,000 × *g* and 4 °C. The cleared lysate was supplemented with CaCl₂ to a final concentration of 2 mM and incubated for 1 h at 4 °C with washed Sepharose 4B Calmodulin beads (Agilent # 214303-52) in a spinning rotor. The beads were washed 5 times with 5 ml of S-CDK binding buffer (40 mM HEPES-KOH pH 7.6, 0.02% NP40 substitute, 10% glycerol, 300 mM KOAc, and 2 mM CaCl₂), and 5 times with 5 ml of S-CDK-TEV buffer (40 mM HEPES-KOH pH 7.6, 0.02% NP40 substitute, 10% glycerol, and 2 mM CaCl₂). S-CDK was then cleaved from the beads by incubating overnight with TEV protease at 4 °C in S-CDK TEV buffer. Released S-CDK was then concentrated in an Amicon Ultra-4 Ultracell 30 kDa centrifugal filter (Merck-Millipore # UFC803024), and injected into a Superdex 200 Increase 10/300 GL column (Cytiva # 15182085) equilibrated in S-CDK GF buffer (40 mM HEPES-KOH pH 7.6, 0.01% NP40 substitute, 10% glycerol, and 300 mM KOAc). Peak fractions were pooled and concentrated in an Amicon Ultra-4 Ultracell 30 kDa centrifugal filter (Merck-Millipore # UFC803024). Aliquots were snap frozen and stored at −80 °C.

**Dpb11.** Dpb11 with a C-terminal 3xFlag-tag was purified from *S. cerevisiae* strain yJY26[7]. Powder was suspended in Dpb11 lysis buffer (25 mM HEPES-KOH pH 7.6, 0.02% NP40 substitute, 10% glycerol, 500 mM KCl, 1 mM EDTA, and 1 mM DTT) supplemented with protease inhibitors. The lysate was cleared in a Beckman-Coulter ultracentrifuge (type Optima L90K with rotor TI45) for 1 h at 235,000 g and 4 °C. The cleared lysate was incubated for 1 h at 4 °C with washed M2 anti-flag affinity beads (Merck-Sigma # A2220; 1–2 mL of beads per 40 mL of lysate) in a spinning rotor. The beads were washed 10 times with 5 ml Dpb11 lysis buffer and Dpb11 was eluted from the beads by incubation with 3xFLAG peptides (Merck-Sigma # F4799). The eluate was dialyzed against Dpb11 dialysis buffer I (25 mM HEPES-KOH pH 7.6, 0.02% NP40 substitute, 10% glycerol, 150 mM KCl, 1 mM EDTA, and 1 mM DTT), passed through a 0.2-micron filter and injected into a 1-ml MonoS 5/50 GL column (GE Healthcare # 17-5168-01) equilibrated in Dpb11 MonoS buffer A (25 mM HEPES-KOH pH 7.6. 0.02% NP40 substitute, 10% glycerol, 150 mM KCl, 1 mM EDTA, and 1 mM DTT). The column was washed with Dpb11 MonoS buffer A, and Dpb11 was eluted from the column in a 20 CV KCl gradient from 150 mM to 1000 mM. Peak fractions were pooled, concentrated in an Amicon Ultra-4 Ultracell 30 kDa centrifugal filter (Merck-Millipore # UFC803024), and injected into a Superdex 200 Increase 10/300 GL column (Cytiva # 15182085) equilibrated in Dpb11 GF buffer (25 mM HEPES-KOH pH 7.6, 0.02% NP40 substitute, 10% glycerol, 300 mM KOAc, 1 mM EDTA, and 1 mM DTT). Peak fractions were pooled and concentrated in an Amicon Ultra-4 Ultracell 30 kDa centrifugal filter (Merck-Millipore # UFC803024). Aliquots were snap frozen and stored at −80 °C.

**Sld2.** Sld2 with a C-terminal 3xFLAG-tag was purified from *S. cerevisiae* strain yTD8[7]. Powder was suspended in Sld2 lysis buffer (25 mM HEPES-KOH pH 7.6, 0.02% NP40 substitute, 10% glycerol, 500 mM KCl, 1 mM EDTA, and 1 mM DTT) supplemented with 0.3 mM PMSF, 7.5 mM benzamidine, 0.5 mM AEBSF, 1 mM leupeptin, 1 mM pepstatin A, and 1 µg/ml aprotinin. The lysate was cleared in a Beckman-Coulter ultracentrifuge (type Optima L90K with rotorTI45) for 1 h at 235,000 g and

4 °C. Solid ammonium sulfate was added to the cleared lysate up to a saturation of 32%. After 15 min of tumbling at 4 °C, the lysate was cleared by centrifugation at 27000 g for 20 min. Then, solid ammonium sulfate was added to the supernatant up to a saturation of 48%. After 15 min of tumbling at 4 °C, the lysate was cleared by centrifugation at 27000 g for 20 min. The pellet was dissolved in Sld2 lysis buffer supplemented with 0.3 mM PMSF, 7.5 mM benzamidine, 0.5 mM AEBSF, 1 mM leupeptin, 1 mM pepstatin A, and 1 µg/ml aprotinin, and incubated for 30 min at 4 °C with washed M2 anti-FLAG beads (Merck-Sigma # A2220; 1–2 mL of beads per 40 mL of lysate) in a spinning rotor. The beads were washed 8 times with 5 ml of Sld2 lysis buffer, tumbled for 10 min at 4 °C with 10 ml of FLAG resuspension buffer (25 mM HEPES-KOH pH 7.6, 0.02% NP40 substitute, 10% glycerol, 500 mM KCl, 1 mM ATP, 10 mM MgOAc, and 1 mM DTT) supplemented with 0.3 mM PMSF, 7.5 mM benzamidine, 0.5 mM AEBSF, 1 mM leupeptin, 1 mM pepstatin A, and 1 µg/ml aprotinin, and 8 more times with 5 ml of Sld2 lysis buffer. Sld2 was eluted from the beads by incubation with 3xFLAG peptides (Merck-Sigma # F4799). The eluate was dialyzed for 45 min at 4 °C against Sld2 dialysis buffer I (25 mM HEPES-KOH pH 7.6, 0.02% NP40 substitute, 10% glycerol, 280 mM KCl, 1 mM EDTA, and 1 mM DTT), passed through a 0.2 micron filter, and injected into a 1-ml HiTrap SPFF column (GE Healthcare # 17-5054-01) equilibrated in Sld2 SPFF buffer A (25 mM HEPES-KOH pH 7.6, 0.02% NP40 substitute, 10% glycerol, 250 mM KCl, 1 mM EDTA, and 1 mM DTT). The column was washed with Sld2 SPFF buffer A, and Sld2 was eluted from the column in a 20-CV KCl gradient from 280 mM to 1000 mM. Peak fractions were pooled, dialyzed for 45 min at 4 °C against Sld2 dialysis buffer II (25 mM HEPES-KOH pH 7.6, 0.02% NP40 substitute, 40% glycerol, 350 mM KCl, 1 mM EDTA, and 1 mM DTT). Aliquots were snap frozen and stored at −80 °C.

**Sld3/7.** Sld3/7 with a C-terminal CTP tag on Sld3 was purified from *S. cerevisiae* strain yTD6[7]. Powder was suspended in Sld3/7 lysis buffer (25 mM HEPES-KOH pH 7.6, 0.02% NP40 substitute, 10% glycerol, 500 mM KCl, 1 mM EDTA, and 1 mM DTT) supplemented with protease inhibitors. The lysate was cleared in a Beckman-Coulter ultracentrifuge (type Optima L90K with rotor TI45) for 1 h at 235,000 g and 4 °C, and incubated for 40 min at 4 °C with washed IgG Sepharose 6 Fast Flow (GE Healthcare cat # 17-0969-01) in a spinning rotor. The beads were washed with 15CV Sld3/7 wash buffer (25 mM HEPES-KOH pH 7.6, 0.02% NP40 substitute, 10% glycerol, 500 mM KCl, 0.5 mM EDTA, and 1 mM DTT), and the protein complex was cleaved from the beads by overnight treatment at 4 °C with TEV protease in Sld3/7 lysis buffer. Sld3/7 was eluted from the column, concentrated in an Amicon Ultra-4 Ultracell 30 kDa centrifugal filter (Merck-Millipore # UFC803024), and injected into a Superdex 200 Increase 10/300 GL column (Cytiva # 15182085) equilibrated in Sld3/7 GF buffer (25 mM hepes-KOH pH 7.6, 0.02% NP40 substitute, 10% glycerol, 500 mM KCl, 1 mM EDTA, and 1 mM DTT). Peak fractions were pooled and concentrated in an Amicon Ultra-4 Ultracell 30 kDa centrifugal filter (Merck-Millipore # UFC803024). Aliquots were snap frozen and stored at −80 °C.

**Cdc45 and Cdc45-iS6.** Cdc45 and Cdc45-iS6: Cdc45 with an internal 2xFLAG tag and Cdc45 with an internal 2xFLAG + S6 tag were purified from S. *cerevisiae* strains yJY13[7] and yDRM2 (this study), respectively. Powder was suspended in Cdc45 lysis buffer (25 mM HEPES-KOH pH 7.6, 10% glycerol, 500 mM KOAc, 1 mM EDTA, and 1 mM DTT) supplemented with protease inhibitors. The lysate was cleared in a Beckman-Coulter ultracentrifuge (type Optima L90K with rotor TI45) for 1 h at 235,000 × *g* and 4 °C. The cleared lysate was incubated for 1 h with washed M2 anti-flag affinity beads (Merck-Sigma # A2220; 1–2 mL of beads per 40 mL of lysate) at 4 °C in a spinning rotor. The beads were washed 10 times with 5 ml of Cdc45 lysis buffer, and Cdc45 was eluted from the beads by incubation with 3xFLAG peptides (Merck-Sigma # F4799). The eluate was dialyzed against Cdc45 dialysis buffer I

(20 mM K phosphate pH 7.6, 10% glycerol, 150 mM KOAc, and 0.5 mM DTT) and injected into a 2-ml hydroxyapatite Bio gel HTP column (Biorad # 130-0420) equilibrated in Cdc45 HTP equilibration buffer (20 mM K phosphate pH 7.6, 10% glycerol, 150 mM KOAc, and 0.5 mM DTT). Cdc45 bound to the hydroxyapatite Bio gel for 45 min at 4 °C with tumbling. The column was then washed with Cdc45 wash buffer A (80 mM K phosphate pH 7.6, 10% glycerol, 150 mM KOAc, and 0.5 mM DTT), and Cdc45 was eluted with Cdc45 HTP elution buffer (250 mM K phosphate pH 7.6, 10% glycerol, 150 mM KOAc and 0.5 mM DTT). Positive fractions were pooled and dialyzed against Cdc45 dialysis buffer I (25 mM HEPES-KOH pH 7.6, 10% glycerol, 300 mM KOAc, 1 mM EDTA, and 1 mM DTT). Finally, Cdc45 was concentrated in an Amicon Ultra-4 Ultracell 30 kDa centrifugal filter (Merck-Millipore # UFC803024). Aliquots were snap frozen and stored at −80 °C.

**RPA.** RPA with a CBP-TEV tag on Rfa1 was purified from *S. cerevisiae* strain yAE31[7]. Powder was suspended in RPA lysis buffer (25 mM Tris-HCl pH 7.2, 10% glycerol, 500 mM NaCl, and 1 mM DTT) supplemented with protease inhibitors. The lysate was cleared in a Beckman-Coulter ultracentrifuge (type Optima L90K with rotorTI45) for 1 h at 235,000×$g$ and 4 °C. The cleared lysate was diluted with an equal volume of RPA dilution buffer I (25 mM Tris-HCl pH 7.2, 10% glycerol, and 1 mM DTT), supplemented with $CaCl_2$ to a final concentration of 2 mM, and incubated for 1 h at 4 °C with washed Sepharose 4B Calmodulin beads (GE Healthcare # 17-0529-01) in a spinning rotor. The beads were washed 10 times with 5 ml of RPA binding buffer (25 mM Tris-HCl pH 7.2, 10% glycerol, 200 mM NaCl, 2 mM $CaCl_2$, and 1 mM DTT), and the protein complex was eluted from the beads with RPA elution buffer (25 mM Tris-HCl pH 7.2, 10% glycerol, 200 mM NaCl, 2 mM EDTA, 2 mM EGTA, and 1 mM DTT). Positive fractions were pooled and diluted with an equal volume of RPA dilution buffer II (25 mM Tris-HCl pH 7.2, 10% glycerol, 1 mM EDTA, and 1 mM DTT) and twice dialyzed for 1 h against RPA dialysis buffer (25 mM Tris-HCl pH 7.2, 10% glycerol, 50 mM NaCl, 1 mM EDTA, and 1 mM DTT). The sample was then loaded onto a 1-ml Hi Trap heparin HP column (GE Healthcare # 17-0406-01) equilibrated with buffer RPA heparin A (25 mM Tris-HCl pH 7.2, 10% glycerol, 50 mM NaCl, 1 mM EDTA, and 1 mM DTT). The column was washed with RPA heparin buffer A, and RPA was eluted from the column in a 30 CV NaCl gradient from 50 mM to 1000 mM. Subsequently, the sample was concentrated in an Amicon Ultra-4 Ultracell 30 kDa centrifugal filter (Merck-Millipore # UFC803024), and injected into a Superdex 200 Increase 10/300 GL column (Cytiva # 15182085) equilibrated in RPA GF buffer (25 mM Tris-HCl pH 7.2, 10% glycerol, 150 mM NaCl, 1 mM EDTA, and 1 mM DTT). Peak fractions were pooled and concentrated in an Amicon Ultra-4 Ultracell 30 kDa centrifugal filter (Merck-Millipore # UFC803024). Aliquots were snap frozen and stored at −80 °C.

**Mcm10.** Mcm10 with a N-terminal 6xHis-tag and a C-terminal 3xFLAG tag (Max Douglas, Francis Crick Institute, unpublished) was purified from *E. coli* BL21-Codon Plus (DE3)-RIL. Cleared lysate was incubated for 1 h at 4 °C with washed M2 anti-flag affinity beads (Merck-Sigma # A2220; 1–2 mL of beads per 40 mL of lysate) in a spinning rotor. The beads were washed 10 times with 5 ml of Mcm10 lysis buffer (25 mM Tris-HCl pH 7.2, 10% glycerol, 500 mM NaCl, and 0.01% NP40 substitute) and 5 times with 5 ml of Mcm10 lysis buffer with 300 mM NaCl (25 mM Tris-HCl pH 7.2, 10% glycerol, 300 mM NaCl, and 0.01% NP40 substitute). Mcm10 was eluted from the beads by incubation with 3xFLAG peptides (Merck-Sigma # F4799) and incubated with Ni-NTA agarose (Qiagen # 30210) for 1 h at 4 °C. The beads were washed 5 times with 5 ml of Mcm10 wash buffer II (25 mM Tris-HCl pH 7.2, 10% glycerol, 500 mM NaCl, and 0,01% NP40 substitute) and 5 times with 5 ml of Mcm10 wash buffer III (25 mM Tris-HCl pH 7.2, 10% glycerol, 500 mM NaCl, 0.01% NP40 substitute, and 20 mM Imidazole). Then the protein complex was eluted from the beads with RPA elution buffer (25 mM Tris-HCl pH 7.2, 10% glycerol, 500 mM NaCl, 0.01%

NP40 substitute, and 200 mM Imidazole). Positive fractions were pooled and dialyzed against Mcm10 dialysis buffer (25 mM HEPES-KOH pH 7.6, 10% glycerol, 200 mM KOAc, 1 mM EDTA, and 0.01% NP40 substitute). After dialysis, Mcm10 was concentrated in an Amicon Ultra-4 Ultracell 30 kDa centrifugal filter (Merck-Millipore # UFC803024). Aliquots were snap frozen and stored at −80 °C.

**S6-dCas9-Halo.** S6-dCas9-Halo with an N-terminal 6xHis-tag (this study) was purified from *E. coli* BL21-Codon Plus (DE3)-RIL. Cleared lysate was incubated with Ni-NTA agarose (Qiagen # 30210) for 1 h at 4 °C in Cas9 lysis buffer (50 mM Na-phosphate pH 7.0, and 300 mM NaCl) supplemented with protease inhibitors. Beads were washed 5 times with 5 ml of dCas9 wash buffer I (50 mM Na phosphate pH 7.0, 300 mM NaCl, and 20 mM imidazole). Then the protein complex was eluted from the beads with dCas9 elution buffer I (50 mM Na phosphate pH 7.0, 300 mM NaCl, and 150 mM imidazole). Positive fractions were pooled and dialyzed against dCas9 dialysis buffer I (50 mM HEPES-KOH pH 7.6, 100 mM KCl, and 1 mM DTT). Then the sample was passed through a 0.2 micron filter and injected into a 1-ml HiTrap SP HP column (GE Healthcare # 17-1151-01) equilibrated in dCas9 SPHP buffer A (50 mM HEPES-KOH pH 7.6, 100 mM KCl, and 1 mM DTT). S6-dCas9 was eluted from the column in a 30 CV KCl gradient from 100 mM to 1000 mM. Positive fractions were concentrated in an Amicon Ultra-4 Ultracell 30 kDa centrifugal filter (Merck Millipore # UFC803024), and injected into a Superose 6 Increase 10/300 GL column (GE Healthcare # 29-0915-96) equilibrated in dCas9 GF buffer (50 mM HEPES-KOH pH 7.6, 150 mM KCl, 20% glycerol, and 1 mM DTT). Peak fractions were pooled and concentrated in an Amicon Ultra-4 Ultracell 30 kDa centrifugal filter (Merck-Millipore # UFC803024). Aliquots were snap frozen and stored at −80 °C.

**Sfp phosphopantetheinyl transferase.** Sfp phosphopantetheinyl transferase with a C-terminal 6xHis-tag (Addgene # 75015) was purified from *E. coli* BL21-Codon Plus (DE3)-RIL. Cleared lysate was incubated with Ni-NTA agarose (Qiagen # 30210) for 30 min at 4 °C in Sfp lysis buffer (20 mM Tris-HCl, pH 7.9, 500 mM NaCl, and 10 mM imidazole) supplemented with protease inhibitors The beads were washed with 100 mL of Sfp wash buffer (20 mM Tris-HCl, pH 7.9, 500 mM NaCl, and 30 mM imidazole), and protein was eluted with Sfp elution buffer (20 mM Tris-HCl, pH 7.9, 500 mM NaCl, and 250 mM imidazole). Positive fractions were pooled and dialyzed overnight against Sfp dialysis buffer I (50 mM HEPES-KOH, pH 7.6, 100 mM KCl, and 50% glycerol). Dialyzed Sfp transferase was then concentrated 12-fold in a 3 kDa Amicon® Ultra-15 Centrifugal Filter Units (Millipore # UFC9003). Aliquots were snap frozen and stored at −80 °C.

### Protein labeling

**dCas9[LD555].** S6-dCas9-Halo was fluorescently labeled by incubating S6-dCas9 with Sfp phosphopantetheinyl transferase and LD555-CoA (Lumidyne Technologies, custom synthesis) in a 1:2:10 molar ratio in dCas9 gel filtration buffer (50 mM HEPES/KOH, pH 7.6, 150 mM KCl, 20% glycerol, and 1 mM DTT) supplemented with 10 mM $MgCl_2$ at RT for 1 h. dCas9[LD555] was separated from unincorporated dye and Sfp phosphopantetheinyl transferase by gel filtration in a Superdex 200 Increase 10/300 GL column (Cytiva # 15182085).

**dCas9[JF646].** Labeling of dCas9-Halo with fluorescent dye JF646 was carried out as previously described[2].

**Mcm2-7[JF646-Mcm3].** Labeling of Mcm2-7[Halo-Mcm3] with fluorescent dye JF646-HaloTag ligand was carried out as previously described[2]. The labeling efficiency was determined to be at least 80% after estimating protein and fluorophore concentrations relative to known standards. Using this method, we obtained a distribution of number of Mcm2-7 complexes per diffraction spots similar to the one obtained in previous

studies in which the labeling efficiency was measured at the single-molecule level[5].

**Cdc45^LD555**. Cdc45-iS6 was fluorescently labeled by incubating Cdc45-iS6 with Sfp phosphopantetheinyl transferase and LD555-CoA (Lumidyne Technologies, custom synthesis) in a 1:1:5 molar ratio in Cdc45 gel filtration buffer (250 K-phosphate, pH 7.6, 150 mM KOAc, 10% glycerol, and 0.5 mM DTT) supplemented with 10 mM MgCl$_2$ at RT for 1 h. Cdc45^LD555 was separated from unincorporated dye and Sfp phosphopantetheinyl transferase by gel filtration in a Superdex 200 Increase 10/300 GL column (Cytiva #15182085). After gel filtration, positive fractions were pooled and concentrated in an Amicon Ultra-4 centrifugal filter Ultracel 30k (Millipore # UFC803024). Labeling efficiency was measured to be $85 \pm 4\%$ (measured value ± instrumental error) by measuring the absorption at 280 nm and 555 nm.

**dCas9^LD555**. S6-dCas9-Halo was fluorescently labeled by incubating S6-dCas9 with Sfp phosphopantetheinyl transferase and LD555-CoA (Lumidyne Technologies, custom synthesis) in a 1:2:10 molar ratio in dCas9 gel filtration buffer (50 mM HEPES/KOH, pH 7.6, 150 mM KCl, 20% glycerol, and 1 mM DTT) supplemented with 10 mM MgCl$_2$ at RT for 1 h. dCas9^LD555 was separated from unincorporated dye and Sfp phosphopantetheinyl transferase by gel filtration in a Superdex 200 Increase 10/300 GL column (Cytiva # 15182085).

**dCas9^JF646**. Labeling of dCas9-Halo with fluorescent dye JF646 was carried out as previously described[2].

### Single-molecule Instrumentation and imaging

**Buffers**
**Buffer A**. 5 mM Tris-HCl pH 7.5, 0.5 mM EDTA, and 1 M NaCl.

**Buffer B**. 10 mM HEPES-KOH pH 7.6, 1 mM EDTA, and 1 M KOAc.

**Buffer C**. 10 mM HEPES-KOH pH 7.6, and 1 mM EDTA.

**Loading buffer**. 25 mM HEPES-KOH pH 7.6, 100 mM K glutamate, 10 mM MgOAc, 0.02% NP40 substitute, 10% glycerol, 2 mM DTT, 100 µg/ml BSA, and 5 mM ATP.

**HSW buffer**. 25 mM HEPES-KOH pH 7.6, 300 mM KCl, 10 mM MgOAc, 0.02% NP40 substitute, 10% glycerol, 1 mM DTT, and 400 µg/ml BSA.

**CMG buffer**. 25 mM HEPES-KOH pH 7.6, 250 mM K glutamate, 10 mM MgOAc, 0.02% NP40 substitute, 10% glycerol, 1 mM DTT, and 400 µg/ml BSA.

**Elution buffer**. CMG buffer supplemented with 10 mM biotin.

**Imaging buffers**. CMG buffer supplemented with 2 mM 1,3,5,7 cyclooctatetraene, 2 mM 4-nitrobenzylalchohol, and 2 mM Trolox.

**DNA functionalization and binding to magnetic beads**. 20 µg of 23.6 kb plasmid pGL50-ARS1 containing a natural ARS1 origin were linearized overnight with AflII (NEB # R0520L). The resulting 4-nt overhangs TTAA at both ends of the linear DNA were functionalized by incorporating desthiobiotinylated dATP (Jena Bioscience # NU-835-Desthiobio) and digoxigenylated dUTP (Jena Bioscience # NU-803-DIGXL) with Klenow Fragment (3′→5′ exo-) (NEB # M0212L); unincorporated nucleotides were removed with Microspin™ S-400 HR spin columns (GE Healthcare # GE27-5140-01). The functionalized DNA was bound overnight at 4 °C to 4 mg of Dynabeads™ M-280 Streptavidin magnetic beads (Invitrogen # 11205D) in Buffer A. After binding, beads were washed twice with Buffer B, twice with Buffer C, and stored at 4 °C in Buffer C. The amount of bound DNA was measured by comparing the concentration of DNA in the supernatant before and after binding, yielding a binding efficiency of 2.3–2.9 mg DNA (~150–190 fmol)/mg beads.

### Hybrid ensemble and single-molecule assay
**Ensemble CMG assembly and activation**. CMG assembly and activation reactions were carried out in two stages: Mcm2-7 loading and phosphorylation, and CMG assembly and activation. Unless otherwise specified, each step of the reaction was conducted at 30 °C with 800 rpm shaking:

**Mcm2-7 loading and phosphorylation**. 1 mg of magnetic DNA-bound magnetic beads was washed with 200 µl of Loading Buffer, and resuspended in 75 µl of Loading Buffer. To load Mcm2-7 hexamers onto the origin-containing DNA, 35.7 nM ORC, 50 nM Cdc6, and 100 nM Mcm2-7/Cdt1 (or Mcm2-7^JF646-Halo-Mcm3/Cdt1 or Mcm2-7^Mcm2(6A)/Cdt1) were incubated with the beads for a total of 30 min, but added to the reaction at 0 min, 5 min and 10 min, respectively. Subsequently, 100 nM DDK was added and the reaction incubated for 30 min. The supernatant was then removed, and the beads were washed once with 200 µl of HSW buffer and once with 200 µl of CMG Buffer.

**CMG assembly and activation**. After washing, beads were resuspended in 50 µl of CMG Buffer supplemented with 5 mM ATP. Then, 50 nM Dpb11, 200 nM GINS, 30 nM Polε, 20 nM S-CDK, 50 nM Cdc45^LD555, 30 nM Sld3/7, 55 nM Sld2, and 10 nM Mcm10 were added to the reaction and incubated for 15 min; For this step, a master mix of all the proteins was made immediately before and incubated on ice. After CMG assembly and activation, the supernatant was removed, and the beads were washed three times with 200 µl of HSW Buffer and once with 100 µl of CMG Buffer. After washing, the assembled DNA-protein complexes were eluted by resuspending the magnetic beads in 200 µl of Elution Buffer, and incubated at RT for 1 h with 800 rpm shaking. The supernatant was then removed and diluted by the addition of 1400 µl of CMG Buffer, and divided into two 700 µl samples for single-molecule imaging.

**Single-molecule imaging**. In general, single-molecule experiments were performed simultaneously on two instruments that combine optical tweezers and confocal microscopy (C-Trap and Q-Trap, LUMICKS); The only exceptions to this are the two-color colocalization experiments and the experiments with the 6 A mutant, which were carried out solely in the C-Trap (LUMICKS). Both instruments use a microfluidic chip with five inlets and one outlet. Three of these channels are injected from the left and used for bead trapping and DNA-protein complex-trapping. The other two channels were used as protein reservoirs and buffer exchange locations (Fig. 1a). Prior to each experiment, the microfluidic chips of both instruments were passivated for at least 30 min with 1 mg/mL bovine serum albumin (BSA, NEB # B9000S) followed by 0.5% Pluronic® F-127 (Sigma # P2443).

In all experiments, the channels contained the following solutions:

**Channel 1**. 2.06 µm anti-digoxigenin coated polystyrene beads (Spherotech # DIGP-20-2) diluted 1:50 in PBS.

**Channel 2**. CMG-containing DNA eluted from magnetic beads.

**Channel 3**. Imaging buffer.

**Channel 4 and 5**. Imaging buffer and imaging buffer supplemented with 5 mM ATP or 5 mM ATPγS (Roche # 11162306001).

Prior to each experiment, the trapping laser power was adjusted to achieve a stiffness of 0.3 pN/nm in both traps[16,41]. Then, individual

DNA-molecules were trapped between two beads in channel 2, and the tethering of single DNA molecules was confirmed by analyzing the force-extension curve[42]. The DNA was then transferred to either channel 4 or channel 5. The distance between both beads was then fixed to achieve a tension of 2 pN, and the DNA was imaged without flow. In all single-color experiments, fluorescent dye LD555 was illuminated with a 561 nm laser at a power of 4 μW as measured at the objective, and the fluorescence was detected on a single-photon counting detector. 2D confocal scans were performed over an area of $160 \times 18$ pixels, which covered the entire DNA stretched at a tension of 2 pN and the edges of both beads. Pixel size was set to $50 \times 50$ nm, illumination time per pixel was set to 0.2 ms, and the frame rate was set to 5 s.

Dual-color experiments were carried out almost identically, with the following differences: 1) fluorescent dyes LD555 and JF646 were simultaneously illuminated with a 561 nm laser at a power of 4 μW and a 638 nm laser at a power of 12.5 μW, and 2) the frame rate was set to 0.7 s. The microscopes output HDF5 files that store the confocal scan data, as well as force data and bead location data monitored during the scan.

## Ensemble assays
### CMG sliding assay
**DNA template generation.** Both 1.4 kb DNA constructs used had one biotinylated end and the same overall sequence containing an ARS1 origin and an HpaII methyltransferase recognition site (CCGG) at the other end. However, only one of the constructs contained a 5-fluoro-2′-deoxy-cytosine within this recognition site to covalently trap the methyltransferase[27,43]. Both constructs were synthesized by PCR using gBlock™ DRM8 (IDT, custom synthesis) as a template and primer pairs DRM_222 and DRM_220 (for the construct without a protein crosslink), or DRM_222 and DRM_218 (for the construct with a protein crosslink). Both reactions were run on a 0.8% agarose gel, and the appropriate bands were purified from the gel and stored at −20 °C.

**DNA:protein crosslink formation and binding to magnetic beads.** 2.5 μg of each DNA construct were incubated at 37 °C overnight with HpaII methyltransferase in a 50:1 protein:DNA molar ratio in CutSmart™ buffer (NEB) supplemented with 10 μM S-adenosylmethionine (NEB #B9003S). Then, each reaction was bound to 1.5 mg of Dynabeads™ M-280 Streptavidin magnetic beads (Invitrogen #11205D) in Buffer A for 1 h at 37 °C and 1000 rpm shaking. After binding, beads were washed twice with Buffer B, twice with Buffer C, and stored at 4 °C in Buffer C.

**CMG sliding assay.** Unless otherwise specified, every step of the reaction was carried out at 30 °C with 1250 rpm shaking. For each condition, 250 μg of DNA-bound magnetic beads were washed with 50 μl of Loading Buffer. Then, 35.7 nM ORC, 50 nM Cdc6, and 100 nM Mcm2-7/Cdt1 (or Mcm2-7$^{Mcm2(6A)}$/Cdt1) were added and incubated with the DNA-bound beads for 30 min; a master mix of all the proteins was made immediately before addition and incubated on ice. Subsequently, 100 nM DDK was added, and the reaction incubated for 30 min. The supernatant was removed, and the beads were washed once with 50 μl of HSW Buffer 2 (25 mM HEPES-KOH pH 7.6, 500 mM NaCl, 10 mM MgOAc, 0.02% NP40 substitute, 10% glycerol, 1 mM DTT, and 400 μg/ml BSA) and once with 50 μl of CMG Buffer. Beads were then resuspended in 50 μl of CMG Buffer supplemented with 5 mM ATP, 50 nM Dpb11, 200 nM GINS, 30 nM Polε, 20 nM S-CDK, 50 nM Cdc45$^{LD555}$, 30 nM Sld3/7, and 55 nM Sld2 and incubated for 5 min; a master mix of all the proteins was made immediately before and incubated on ice. After CMG assembly, the supernatant was removed, and the beads were washed once with 50 μl of HSW Buffer 2 and once with 50 μl of CMG Buffer. After washing, beads were resuspended in 110 μl of HSW Buffer (containing 300 mM KCl) with or without 5 mM

ATPγS, and incubated at 30 °C with 1250 rpm shaking. At the indicated time points, a 20 μl sample was taken from each reaction, and beads were washed with 40 μl of CMG buffer; beads were then resuspended in 15 μl of MNase Elution Buffer (45 mM HEPES-KOH pH 7.6, 300 mM KOAc, 5 mM MgOAc, 2 mM CaCl₂, and 10% glycerol) supplemented with 0.45 μl of Microccocal nuclease (NEB # M0247S), and incubated at 30 °C for 2 min without shaking. The supernatant was then collected and run on a 4–12% Bis-Tris polyacrylamide gel. To monitor the amount of bound fluorescent Cdc45, gels were scanned with a green laser on an Amersham Typhoon. Densitometry was performed on ImageJ.

**Unwinding assay.** All ensemble unwinding assays were carried out as previously described[10].

## Data analysis
**Software and code.** We used Python 3.8 with several libraries for image processing. We used the Laplacian of Gaussian detector from Python's "scipy" for spot detection. We used the Linear Assignment Problem method[44] and the "scipy" solver "linear_sum_assignment" to do spot tracking. Bleaching trace analysis was done with the "ruptures" library. A full list of the exact python libraries and their versions: numpy==1.19.5; matplotlib==3.2.2; lumicks-pylake==0.7.1; streamlit==0.74.1; scipy==1.6.1; scikit-image==0.16.2; scikit-learn==0.23.1; pyyaml==5.3.1; pandas==1.0.5; pillow==7.2.0; tifffile==2021.1.11; jupyterlab==2.1.5; notebook==6.0.3; ruptures==1.1.6; pykalman==0.9.5.

**Overview of data analysis.** After taking confocal scans, the resulting raw image data was processed to generate a table containing the spot detections in each frame. These spot detections are connected between frames to produce traces that contain location and intensity information over time.

During the subsequent motion analysis, we fit linear segments to each location trace. The resulting velocities are used to determine whether a trace is static or not. Finally, we determine the type of motion of each non-static trace using anomalous diffusion analysis.

### Acquiring trace data from raw images
**Spot detection and tracking.** For spot detection we use the scikit-image implementation of a Laplacian of Gaussian (LoG) blob detector[45]. We set the detection radius $r_{LoG}$ to 5 pixels (250 nm); the LoG sigma parameter is then given by $\sigma_{LoG} = r_{LoG} / \sqrt{2}$. The detection threshold is set to 0.5 ADU/pixel[2]. Detected spots are localized with subpixel resolution by performing Gaussian profile fits on spot intensity projections in both x- and y-directions. For frame-by-frame tracking of the spots, we use our own implementation of the Linear Assignment Problem (LAP) framework[44] with a maximum spot linking distance of 6 pixels (300 nm) and a maximum frame gap of 3 frames (15 s). Spots are considered colocalized if they are less than 2 pixels (100 nm) apart. Spot intensities are given by the total photon count within the detection radius.

**Location and fluorophore intensity calibration.** We use the fluorescent dCas9 data (Supplementary Fig. 3j) to calibrate spot locations and expected fluorophore bleaching step sizes. Because the location of the dCas9 on the DNA is known, a pixel-to-base-pair map can be made for the confocal images by mapping the mean location of all dCas9 spots (on the left and the right side of the DNA) to the corresponding locations in base pairs. Moreover, because dCas9 is labeled with the same fluorophores as the fluorescent proteins used in the CMG experiments, and because dCas9 spots contain one dCas9 molecule, we can find the fluorophore bleaching step size mean $\mu_{\Delta I}$ and standard deviation $\sigma_{\Delta I}$. The minimum bleaching step size, needed for bleaching step fitting, is set to $\Delta I_{min} \leq \mu_{\Delta I} - 2\,\sigma_{\Delta I}$ to capture at least 95% of all bleaching events.

**Determination of number of fluorophores per diffraction-limited spot.** To determine the number of fluorescently labeled proteins within each diffraction-limited spot, we count the number of photobleaching step within each spot. For experiments with multiple laser colors (in this case red ($r$) and green ($g$)), we first correct spot intensities for crosstalk by using the equation $I_{r,\ corrected} = I_r - I_g \cdot \mu_{\Delta I,\ r\ (crosstalk)} / \mu_{\Delta I,\ g}$. Then, we fit bleaching traces to a piecewise constant function using Change-Point Analysis (CPA) (we use a Python implementation called 'ruptures'[46]). We use an L2 cost function to detect mean-shifts in the signal, with a minimum segment length of 2 and a penalty term of $\Delta I_{min}2$. If any steps smaller than $\Delta I_{min}$ are detected, these are pruned starting at the smallest step, until only steps larger than $\Delta I_{min}$ remain.

**Data filtering.** The resulting data table of traces with number of fluorescent proteins per spot was filtered in order to reduce noise, outliers, and data that is not suitable for further motion analysis:

(1) While the distance between the optical traps is constant, the force between the traps can fluctuate; jumps in the force signal could indicate, for instance, DNA 'slipping' from the beads, or a protein aggregate landing on a bead, which makes the location signal inaccurate. Hence, if the force signal exhibits a jump larger than $2\sigma_F$ after fitting with CPA, where $\sigma_F = 0.1$ pN is the force fluctuation of a clean trace, only the part of the trace before that jump is used for motion analysis.

(2) Diffraction-limited spots containing more than 5 fluorescent proteins, likely aggregates, are filtered out.

(3) Any traces starting or ending within 1 kbp from a bead are filtered out to prevent any proteins likely stuck to a bead from entering the dataset.

(4) Any traces starting after frame 3 are also filtered away because we do not expect any fluorescent protein to land on the DNA during the scan.

(5) The last frame of each trace is omitted for motion analysis because photobleaching often happens while that frame is being taken, resulting in a distorted spot with an incorrect position.

(6) Finally, in order to perform reliable motion analysis, only traces with a length of 14 frames or more are retained and used for motion analysis (Supplementary Fig. 6c).

**Positional analysis.** In all positional plots, we report the average position of the first three frames of each trace as the initial position of CMG. The bin size of the initial position histograms was set to 700 bp to be close to the diffraction limit while having the ARS1 origin positioned near the center of its corresponding bin.

**Motion analysis**

**Trace segment fitting.** To reduce noise before we fit segments to each trace, we first apply a Kalman filter with expectation-maximization (using the pykalman Python library https://github.com/pykalman/pykalman). Then, we fit linear segments with CPA[46], this time using a linear cost function to fit a multiple linear regression model to the trace. The minimum segment size is set to 3 and the penalty term is set to 0.3; halving or doubling the penalty term does not give a significant change in results, showing that the results are robust. After this procedure, each spot detection has associated with it a CPA-fitted velocity $v_{CPA}$.

The CPA fit makes sense for static traces and traces exhibiting piecewise linear motion, but not for spots undergoing diffusive motion. For the latter, we expect to see CPA segments with randomly alternating directions, and random velocities from some distribution with a variance dictated by the diffusion constant. A detailed description of the analysis of diffusive spots can be found below.

**Motion calibration.** The distribution of $v_{CPA}$ for fluorescent dCas9 gives us two values to calibrate the motion analysis. Firstly, the mean velocity $\mu_v = 0.38$ bp/s provides a drift correction value. Secondly, the standard deviation $\sigma_v = 0.40$ bp/s gives us a cutoff value to determine whether a diffraction-limited spot is static or not; we set this cut-off at the conservative value of $5\sigma_v = 2.0$ bp/s.

Another value we need for further analysis is the location measurement error $\sigma_x$. This error is given by the standard deviation of detected dCas9 locations around their mean, after drift correction, which is found to be $\sigma_x = 72$ bp ($\approx 24$ nm) (Supplementary Fig. 3e).

**Anomalous diffusion analysis.** First, we correct each trace for drift with $x_{corrected}(t) = x(t) - t \cdot \mu_v$. Then we use mean squared displacement (MSD) analysis[26] to fit an anomalous diffusion exponent $\alpha$, which characterizes the motion type of each mobile trace. The MSD has the form:

$$MSD(\tau) = D_\alpha \tau^\alpha + 2\sigma_x^2 \qquad (1)$$

where $D_\alpha$ is the anomalous diffusion constant and $\tau$ is the lag time. For spots undergoing confined diffusion, $\alpha \ll 1$; for freely diffusive spots $\alpha \approx 1$, and for traces exhibiting unidirectional motion $\alpha \gg 1$. The fit is performed through the logarithm of the measurement error corrected MSD:

$$\log(MSD(\tau) - 2\,\sigma_x^2) = \log(D_\alpha) + \alpha\,\log(\tau) \qquad (2)$$

We use least squares to fit up to a maximum lag time $\tau_M$ of 33% of the total length of the trace, with a minimum $\tau_M$ of 5 frames and a maximum of 50 frames. The value of $\alpha$ is constrained between 0 and 2. The trace is then placed into one of three categories using the fitted value of $\alpha$, with confined diffusive spots $0 \le \alpha < 0.5$, freely diffusive spots $0.5 \le \alpha < 1.5$, and unidirectionally moving spots $1.5 \le \alpha \le 2$. Because we expect populations around $\alpha \approx 1$ and $\alpha \approx 2$, we need the error in alpha $\sigma_\alpha$ to be <0.5 in order to ensure statistically significant results.

**Calculation of diffusion coefficients.** For traces that are found to be diffusive, we calculate the diffusion coefficient by redoing the MSD fit, setting $\alpha = 1$, and using a previously published appropriate range of delay times[47].

**Anomalous diffusion exponent error determination.** In order to study the error in $\alpha$ as a function of minimum trace length, we have run the same analysis on 512 simulated diffusive traces and 512 simulated traces with a constant speed (with $\alpha = 1$ and $\alpha = 2$, respectively), with representative values for the diffusion constant $D = 1.5 \times 10^{-3}$ kb$^2$/s and speed $v = 5$ bp/s. We use the experimentally determined measurement error ($\sigma_x = 72$ bp) and mean fluorophore lifetime (25 frames). These simulations show that we need a minimum trace length of 14 frames for the error in alpha, $\sigma_\alpha$, to be <0.5, justified by the motion classification cutoffs discussed above.

In all plots we use experimental means and standard deviations whenever possible. On population bar plots we use the statistical error, i.e., the standard error of proportion, given by $\sqrt{p(1-p)/n}$, with $p$ the measured proportion and $n$ the sample size.

**Bin size selections.** In general, the bin size of all the histograms in this manuscript were chosen to be larger but in the order of magnitude of the error of the random variable being displayed. Specifically:

- The bin size of the initial position histograms was set to 700 bp to be close to the diffraction limit while having the ARS1 origin positioned near the center of its corresponding bin.
- The bin size of the absolute instantaneous velocities histograms was set to 2.5 bp/s, which is ~6 X the velocity noise (Fig. 2a inset).
- The bin size of the absolute mean velocities histograms was set to 1.0 bp/s, which is ~ 2 X the velocity noise (Fig. 2a inset).

- The bin size of instantaneous velocities from CPA histograms was set to 1.0 bp/s, which is ~ 2X the velocity noise (Fig. 2a inset).
- The bin size of the processivities histograms was set to 0.5 kbp, which is ~ 7X the location error (Supplementary Fig. 3e).
- The bin sizes of the histograms of step sizes and location errors of dCas9 (Supplementary Fig. 3a, b, e) are irrelevant because we only use the means and the standard deviations of the underlying distribution for our analysis.
- The bin sizes of the histograms of anomalous diffusion exponents $\alpha$ are set to 0.25, which is ~ 1/2 the error in $\alpha$ (Supplementary Fig. 6c). These histograms, however, were only an intermediate in our analysis. In the final analysis, a bin size of 0.5 ~ the error in $\alpha$ (Supplementary Fig. 6c, Methods) was used to classify motion types.

## Reporting summary

Further information on research design is available in the Nature Portfolio Reporting Summary linked to this article.

## Data availability

Source data are provided with this paper. The data supporting the findings of this study are available from the corresponding authors upon request. Raw and processed ensemble and single-molecule data generated in this study have been deposited in the 4TU.ResearchData repository and can be found at https://doi.org/10.4121/19948253. The repository contains a table with an overview of experiments, spot position and intensity tables sorted by experimental condition and by setup, filtered spot tracking tables, with connected spot detections for each frame in each scan, each row has a scan_id and trace_id, motion analysis summary tables, containing motion information for each trace and example TIFF files. The scan_id, trace_id fields link this table to the table containing the full trace information. Source data are provided with this paper.

## Code availability

All the code used in the current study is available at https://gitlab.tudelft.nl/nynke-dekker-lab/public/cmg-activation.

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

## Acknowledgements

We thank Anne Early, Lucy Drury, and Max Douglas for providing the yeast strains for the overexpression of unlabeled proteins, Jacob Lewis for providing us with purified M.HpaII methyltransferase, and N.D. lab members Anuj Kumar, Katinka Ligthart, and Julien Gros for assistance in purifying loading proteins and/or DDK. We also thank Kaley McCluskey, Dorian Mikolajczak, Joseph Yeeles, Jacob Lewis, Alessandro Costa, Hasan Yardimci and Taekjip Ha for scientific discussions. D.R.M. acknowledges funding from a Boehringer Ingelheim Fonds PhD Fellowship, and N.D. acknowledges funding from the Netherlands Organisation for Scientific Research (NWO) through Top grant 714.017.002, and from the European Research Council through an Advanced Grant (REPLICHROMA; grant number 789267).

## Author contributions

D.R.M., H.S., J.D., and N.D. conceived the study, and N.D. supervised the study. D.R.M. developed the ensemble hybrid and single-molecule study with input from H.S., J.D., and N.D.; D.R.M. carried out the ensemble biochemical assays and the did ensemble part of the hybrid ensemble and single-molecule assays. D.R.M. and H.S. acquired the single-molecule data. J.D. provided the strains for the overexpression of unlabeled proteins and advised on biochemical conditions. D.R.M. and T.v.L. carried out the cloning. D.R.M. generated the strains for the overexpression of labeled proteins. D.R.M. and T.v.L. purified the proteins. D.R.M. and H.S. labeled the proteins. E.v.V. designed and wrote data analysis routines with input from all authors as well as conceived and performed the simulations. D.R.M. and E.v.V. performed the data analysis. B.S. contributed to project planning and instrument acquisition. All authors were involved in the discussion of the data. D.R.M. and N.D. wrote the manuscript with input from all the authors.

## Competing interests

The authors declare no competing interests.
