## [Peer Review File · Nature Communications]

Nucleotide binding halts diffusion of the eukaryotic replicative helicase during activationEditorial Note: This manuscript has been previously reviewed at another journal that is not operating a transparent peer review scheme. This document only contains reviewer comments and rebuttal letters for versions considered at *Nature Communications*. Mentions of the other journal have been redacted.

REVIEWER COMMENTS

Reviewer #1 (Remarks to the Author):

Most of my points were addressed to a satisfactory level. The manuscript is improved, and the better integration of previous literature gives a better context for the current development. The change of the model to reflect surrounding of dsDNA and ssDNA versus melted and extruded states is also beneficial.

Nevertheless, the authors would greatly benefit from not excluding completely the presence of some heterogeneity. To clarify, since CMG initial recruitment, assembly and activation cannot be directly monitored in single-molecule, there is no direct evidence that all CMG complexes were assembled at the origin. In turn, it is possible that at least a fraction of CMG was matured elsewhere on the DNA in line with the authors reply: 'Furthermore, we cannot fully exclude that some of the CMGs mature elsewhere on the DNA than the origin, as the spatial distribution of the MCM double hexamers that subsequently mature into CMGs is also broadly distributed about the origin (Sánchez et al., Nat. Commun. 2021)'. This fraction that is not recruited at the origin, may have initial associated with the DNA through different mechanisms with different degrees of specificity, and further it may display different characteristics through the subsequent steps. Some of this elsewhere-matured CMGs may be retained and analysed in single-molecule. Since the single-molecule imaging is started at a much later stage, there is no possible way to know the initial distribution of CMG maturation points that produced the observed distribution. Therefore, the authors do not have the necessary experiments to directly show that all the CMG molecules and their subsequent investigated parameters were matured at the origin during the ensemble biochemistry phase. The CMG molecules that matured elsewhere than at the origin may have intrinsically different behaviour. Therefore, I believe that it is critical that the authors include data similar to the one in Figure 1 for a DNA construct that does not include the origin, and a direct comparison of the position distribution (at the beginning of single-molecule imaging) and yield of CMG on the DNA in the presence and absence of the origin in the DNA construct, using exactly the same methodology described in Figure 1. If no very clear origin-specificity is found, then the results should be discussed in a context where CMG matured at the origin and CMG matured elsewhere on the DNA may exhibit different downstream behavior and thus the existence of heterogeneity in assembly.

Reviewer #2 (Remarks to the Author):

This revised manuscript describes a series of single-molecule (SM) and ensemble experiments addressing the function of CMG helicases assembled on DNA from licensed origins. The major take home point of the manuscript is that the ATP bound state of CMG influences its ability to diffuse on DNA. The authors also make conclusions about the rate of CMG directional movement on the DNA in the presence of ATP, that agrees with previous SM studies of pre-assembled CMG molecules. The major addition to the paper is an ensemble assay supporting the authors conclusions based on the SM studies. These studies also included a mutant that does not allow initial DNA melting during CMG activation. Although this experiment leads to the conclusion that DNA melting is not responsible for ATP-dependent diffusion that they observe, it is unfortunate that they did not do this experiment at the SM level. While the authors focus on the claim that these studies are the "first fully in vitro reconstituted single-molecule motion quantification of replicative helicases assembled at an origin of replication", it is noteworthy that the most novel conclusions of the paper are confirmed in an ensemble assay.

Overall, this is a modestly improved manuscript that makes an interesting observation about the engagement of CMG with DNA in the presence of ATP but does not establish the importance of this observation in the context of helicase activation. ¹¹SEP

Specific points.

It is disappointing that the authors did not choose to extend their analysis to ADP. In their response they suggest that ATP-gamma-S (ATPgS) provides some similarity to ADP but they also use ATPgS as an equivalent to ATP bound molecules. Since this seems like a relatively simple addition (a single repeat of the assay would be sufficient), it is surprising that they chose not to do it and certainly lessens the biological relevance of the studies as AAA+ ATPases spend much more time in ATP and ADP bound states than in the no-ATP state.

Given that the authors did not measure the percent labeling of Cdc45 at a SM level and acknowledge the issues with precise measurement of percent labeling in an ensemble setting, they should acknowledge this concern in the main text and include a description of these issues in the methods. It is very likely that lower levels of labeling than they report are a major contributor to the few examples of two CMGs on a DNA.

Reviewer #3 (Remarks to the Author):

This manuscript and review comments were transferred from [redacted]. The authors have addressed all the concerns I raised with the previous submission. Most importantly, they carried out the critical experiment I suggested. The experiment with the Mcm2 6A mutant (which does not melt or extrude DNA) revealed that their observations do not report on the DNA-melted state. The authors have adjusted the model presented in the previous submission, to reflect these new findings.

The work represents a significant technological advancement, introducing the double functionalization of DNA ends with two orthogonal attachment types, the use of dCas9 for calibration in the C-Trap, as well as the first single-molecule observation of CMG assembled from origins.

The observations of the effect of the presence of nucleotides on the behaviour of CMG could reflect molecular pathways at play at the origin or at stalled replication forks. At this stage, it is unclear if this is indeed the case. Nonetheless, combined with the technological advancements, I think this work will be of interest to the DNA-replication field.

We greatly appreciate the fact that all of our three original referees agreed to have a second look at our manuscript, particularly given the busy holiday season. We are happy to see that they are satisfied with the improvements that we made in the manuscript revision (apart from an oversight by Referee 2 that we have pointed out below). The remaining points are also addressed below. Changes to the manuscript and its supplementary information are shown in **green highlights** in those texts.

Kind regards, the Authors

Reviewer #1 (Remarks to the Author):

Most of my points were addressed to a satisfactory level. The manuscript is improved, and the better integration of previous literature gives a better context for the current development. The change of the model to reflect surrounding of dsDNA and ssDNA versus melted and extruded states is also beneficial.

We thank Referee 1 for his/her positive assessment of the improvements that we made during the revision process. This is much appreciated.

Nevertheless, the authors would greatly benefit from not excluding completely the presence of some heterogeneity. To clarify, since CMG initial recruitment, assembly and activation cannot be directly monitored in single-molecule, there is no direct evidence that all CMG complexes were assembled at the origin. In turn, it is possible that at least a fraction of CMG was matured elsewhere on the DNA in line with the authors reply: 'Furthermore, we cannot fully exclude that some of the CMGs mature elsewhere on the DNA than the origin, as the spatial distribution of the MCM double hexamers that subsequently mature into CMGs is also broadly distributed about the origin (Sánchez et al., Nat. Commun. 2021)'. This fraction that is not recruited at the origin, may have initial associated with the DNA through different mechanisms with different degrees of specificity, and further it may display different characteristics through the subsequent steps. Some of this elsewhere-matured CMGs may be retained and analysed in single-molecule. Since the single-molecule imaging is started at a much later stage, there is no possible way to know the initial distribution of CMG maturation points that produced the observed distribution. Therefore, the authors do not have the necessary experiments to directly show that all the CMG molecules and their subsequent investigated parameters were matured at the origin during the ensemble biochemistry phase. The CMG molecules that matured elsewhere than at the origin may have intrinsically different behaviour. Therefore, I believe that it is critical that the authors include data similar to the one in Figure 1 for a DNA construct that does not include the origin, and a direct comparison of the position distribution (at the beginning of single-molecule imaging) and yield of CMG on the DNA in the presence and absence of the origin in the DNA construct, using exactly the same methodology described in Figure 1. If no very clear origin-specificity is found, then the results should be discussed in a context where CMG matured at the origin and CMG matured elsewhere on the DNA may exhibit different downstream behavior and thus the existence of heterogeneity in assembly.

We interpret Referee 1's concerns as two-fold:

1. Can we prove that (much of) CMG formation occurs in an origin-specific manner?
2. Do the dynamics differ for CMG matured at the origin as compared to CMG matured elsewhere?

To address the first concern, we note that in *S. cerevisiae*, as well as in closely related species of yeast, there are DNA sequences known as origins for which in particular the origin recognition complex (ORC) has an established preference. We have confirmed this sequence preference of ORC in our published single-molecule experiments (Sánchez et al., Nat Commun 2021), both for ORC incubated with DNA in the single-molecule flowcell and for ORC pre-incubated with DNA prior to introduction into the single-molecule flowcell (a protocol similar to the approach in the present manuscript). The sequence preference is not absolute, however; as shown in that work, ORC will also bind non-origin sequences, just with lower probability than origin sequences. We also note that for replisome components loaded temporally downstream of ORC, the sequence specificity of their spatial positioning decreases; for example, the Mcm2-7 double hexamer is diffusive when stably loaded onto dsDNA (Remus et al., Cell 2009). This was also confirmed in our published single-molecule experiments (Sánchez et al., Nat Commun 2021), which show a position histogram for Mcm2-7 complexes (generated by pre-incubating loading factors with DNA) that, while centered about the origin, is quite broad. Note that the Mcm2-7 position histograms in Figure 1h,i of the present manuscript recapitulate this previously published behavior. Already, that means that for any given Mcm2-7 hexamer that we observe, we cannot prove whether it originated at the origin or not. Furthermore, *in vivo* Mcm2-

7 loading and maturation into CMG occur in different cell cycle stages (Bell and Labib, Genetics, 2016). Therefore, as Mcm2-7 double hexamers diffuse on dsDNA (Remus et al., Cell 2009; Sánchez, et al., Nature Commun, 2021), it is unlikely that they will be *exactly* at the origin of replication sequences (which are quite short) when they mature into CMG.

The histogram of CMG positions (observed in the single-molecule flowcell following our 15-minute activation in bulk, Figure 1d) in the present manuscript is similarly broadly centered about the origin (with the bin containing the origin being the highest by a bit). This is not at all surprising given the likelihood of finding Mcm2-7 at positions other than the origin. In other words, it is quite possible that some of the CMG will have formed at some distance away from the origin. Our histogram of CMG positions may have broadened further still due to motion of CMG during the ensemble phase of the experiment. This broad histogram of CMG positions is furthermore fully consistent with published ensemble experiments that confirm that CMG maturation need not necessarily occur at the origin in *S. cerevisiae*, for example:

- Mcm2-7 loaded non-specifically (i.e. away from an origin) works in replication, not only for higher eukaryotes but also in *S. cerevisiae* (e.g. Bogenschutz et al., Plos One 2014; Dershowitz et al., Mol Cell Biol 2007; Gros et al., EMBO J 2014; On et al., EMBO J 2014; Takeda et al., Genes Dev 2005).
- It has been shown that if RNA polymerase pushes *S. cerevisiae* MCM double hexamers to a non-origin location, they mature into full CMG, become activated and fire perfectly well (Gros et al., Mol Cell 2015).

Similar flexibility of CMG maturation relative to DNA sequence is standard in higher eukaryotes, which have no known sequence specificity anywhere in the establishment of the replisome.

In summary, we consider that our histogram of CMG positions reflects the previously established origin preference of the ORC protein together with the previously established behavior of downstream events. Collecting data similar to that of Figure 1d but on a DNA without a replication origin as suggested by Referee 1 might slightly decrease the height of the bin containing the origin, but we do not consider that this will be very informative.

To address the second concern, again based on the published ensemble experiments cited above, there is no obvious physiological reason to expect significant differences at the single-molecule level between the dynamics for CMGs matured at the origin versus CMGs matured elsewhere. In our current datasets of CMG dynamics (Figure 2b,c of the manuscript), we find that CMG spots are broadly centered about the origin at time 0 of the single-molecule experiments for all conditions tested (Figure R1).

Figure R1. Initial position distributions in the single-molecule experiments of the CMG spots whose dynamics were analyzed in Figure 2b,c of our manuscript. (a,b) The initial position distribution of CMG spots in the single-molecule experiments that were then characterized as mobile, static, respectively in a buffer containing ATP. Total of 43 traces analyzed in both panels together (same statistics as Figure 2b of our manuscript). **(c,d)** The initial position distribution of CMG spots in the single-molecule experiments that were then characterized as mobile, static, respectively in a buffer without ATP. Total of 36 traces analyzed in both panels together (same statistics as Figure 2c of our manuscript).

Further investigation aimed at investigating more precisely whether CMG dynamics differ between CMGs matured at the origin versus elsewhere would be very challenging in our experiments, because:

- It is not possible for us to prove where exactly any given CMG matured in the first place, as the single-molecule experiments do not have this resolution. Our uncertainty in exactly pinpointing the genomic location of a CMG in the single-molecule experiments is ~750 bp wide (bin size in Figure 1d), which is quite a bit wider than the width of the ARS1 origin). Thus, even for CMGs initially localized in the bin containing the origin, we cannot prove that they matured at the origin. And for CMGs initially localized in other bins (which is the majority of the data), these CMGs either matured at the origin and moved away from it, or perhaps they matured elsewhere on the DNA.
- In other words, whilst in the case of a DNA without a replication origin we would be sure that CMGs could not have matured at the origin, in the experiments that we have already performed we cannot assign their starting point (note that this is the same issue that we pointed out for Mcm2-7 on the previous page: “Already, that means that for any given Mcm2-7 hexamer that we observe, we cannot prove whether it originated at the origin or not.”). Thus, a comparison of the dynamics based on an origin- or non-origin-based starting point will be very difficult.

Thus, we would certainly not be able to conduct a meaningful discussion on the influence of how the dynamics for CMGs matured at the origin versus elsewhere on the basis of any data that we could currently collect.

Reviewer #2 (Remarks to the Author)

This revised manuscript describes a series of single-molecule (SM) and ensemble experiments addressing the function of CMG helicases assembled on DNA from licensed origins. The major take home point of the manuscript is that the ATP bound state of CMG influences its ability to diffuse on DNA. The authors also make conclusions about the rate of CMG directional movement on the DNA in the presence of ATP, that agrees with previous SM studies of pre-assembled CMG molecules. The major addition to the paper is an ensemble assay supporting the authors conclusions based on the SM studies. These studies also included a mutant that does not allow initial DNA melting during CMG activation. Although this experiment leads to the conclusion that DNA melting is not responsible for ATP-dependent diffusion that they observe, it is unfortunate that they did not do this experiment at the SM level. While the authors focus on the claim that these studies are the “first fully in vitro reconstituted single-molecule motion quantification of replicative helicases assembled at an origin of replication”, it is noteworthy that the most novel conclusions of the paper are confirmed in an ensemble assay.

Referee 2 has regrettably overlooked our new single-molecule experiments with the Mcm2 6A mutant, which are described in the Results (lines 246-259) and shown in Extended Data Figure 9. They were clearly mentioned in our point-by-point rebuttal (on p.2, twice on p.7, on p.8, and on p.9). Thus, the statement by the referee “it is unfortunate that they did not do this experiment at the SM level” is incorrect.

The follow-up statement by Referee 2: “it is noteworthy that the most novel conclusions of the paper are confirmed in an ensemble assay” is therefore also incorrect. Our observations about the nucleotide dependence of CMG dynamics were first made at the single-molecule level; we performed ensemble experiments for two reasons: (1) for independent – but less quantitative – verification of the phenomena that we observed at the single-molecule level; (2) to verify hypotheses generated by the single-molecule experiments that required leaving out Mcm10 (which is more accessible at the ensemble level). We consider that our manuscript constitutes a very nice example of how single-molecule and ensemble experiments can jointly lead to novel insights.

Overall, this is a modestly improved manuscript that makes an interesting observation about the engagement of CMG with DNA in the presence of ATP but does not establish the importance of this observation in the context of helicase activation.

We thank Referee 2 for appreciating the interest of our observations of CMG engagement with DNA. We hope that Referee 2 additionally appreciates the new single-molecule experiments with the 6A mutant that were regrettably overlooked as described above. Regarding the comment on the context of helicase activation, please see our response to the next point.

Specific points.

It is disappointing that the authors did not choose to extend their analysis to ADP. In their response they suggest that ATP-gamma-S (ATPgS) provides some similarity to ADP but they also use ATPgS as an equivalent to ATP bound molecules. Since this seems like a relatively simple addition (a single repeat of the assay would be sufficient), it is surprising that they chose not to do it and certainly lessens the biological relevance of the studies as AAA+ ATPases spend much more time in ATP and ADP bound states than in the no-ATP state.

In our experiments, we have examined CMG dynamics (a) in single-molecule experiments in the presence of Mcm10 (for the cases of no nucleotide, ATPgS, and ATP) and (b) in ensemble experiments in the absence of Mcm10 (for the cases of no nucleotide and ATPgS). Based on our findings from these experiments, which demonstrate that CMG is diffusive on dsDNA in the absence of nucleotide, we propose in the Discussion that *in vivo*, nuclear ATP binding helps to maintain CMG in the vicinity of replication origins, thus providing context to helicase activation. Furthermore, binding of CMG to the DNA within its central channel is a critical step in the initial DNA melting required for strand extrusion, which was previously shown to be driven by ATP binding energy (Douglas, Nature 2018).

We have not considered it essential to the findings of our manuscript to specifically address the influence of ADP on the mobility of CMG on dsDNA in our experiments. *In vivo*, CMG would not likely be bound to ADP while encircling dsDNA, because ATP hydrolysis by CMG in the presence of Mcm10 (Douglas et al., Nature 2018) results in ejection of the lagging strand from CMG. Hence we do not see a strong physiological motivation for probing the mobility of CMG on dsDNA in the presence of ADP. CMG can be observed (partially) bound to ADP while encircling ssDNA (see Eichhoff et al., Cell Reports, 2019), in the context of its normal ATP hydrolysis-powered translocation along ssDNA.

We hope therefore that the manuscript and its findings as presented in the current form are acceptable.

Given that the authors did not measure the percent labeling of Cdc45 at a SM level and acknowledge the issues with precise measurement of percent labeling in an ensemble setting, they should acknowledge this concern in the main text and include a description of these issues in the methods. It is very likely that lower levels of labeling than they report are a major contributor to the few examples of two CMGs on a DNA.

It seems unlikely to us that an overestimation of our labeling efficiency would be a major contributor relative to the other factors that we already described in the main text. One reason that we consider it unlikely is that we can compare our published data on Mcm2-7 (Sánchez et al., Nat Commun 2021) obtained using a labeling efficiency of ~80% (estimated according to the same methodology that is described in the Methods of the present manuscript) to published data on Mcm2-7 (Ticau et al., Cell 2015) obtained using a labeling efficiency of 79% (estimated according to the single-molecule methodology suggested by Referee 2). Histograms from Sánchez et al., Nat Commun 2021 show 65-73% Mcm2-7 single hexamers and 24-27% Mcm2-7 double hexamers (the range comes from including higher stoichiometries or not), whereas the table from Ticau et al. shows 68% Mcm2-7 single hexamers and 32% Mcm2-7 double hexamers. Based on the similarity of the single-molecule measurements, it follows that our methodology for estimating labeling efficiency is also quite accurate.

Changes to the manuscript: To reflect this discussion in the present manuscript, we have modified the main text where we mention labeling efficiency; added a reference to the Methods to add this point of the main text; and expanded the Methods to make the above-mentioned point that the nearly identical labeling efficiencies that we obtain for Mcm2-7 using our bulk methodology (Sánchez et al., Nat Commun 2021) and that Ticau et al., Cell 2015 obtain using their single-molecule methodology give similar results for MCM stoichiometry (Sánchez et al., Nat Commun 2021 and Ticau et al., Cell 2015), suggesting that our bulk methodology provides a reliable estimation of the labeling efficiency.

Reviewer #3 (Remarks to the Author)

This manuscript and review comments were transferred from [redacted]. The authors have addressed all the concerns I raised with the previous submission. Most importantly, they carried out the critical experiment I suggested. The experiment with the Mcm2 6A mutant (which does not melt or extrude DNA) revealed that their observations do not report on the DNA-melted state. The authors have adjusted the model presented in the previous submission, to reflect these new findings.

The work represents a significant technological advancement, introducing the double functionalization of DNA ends with two orthogonal attachment types, the use of dCas9 for calibration in the C-Trap, as well as the first single-molecule observation of CMG assembled from origins.

The observations of the effect of the presence of nucleotides on the behaviour of CMG could reflect molecular pathways at play at the origin or at stalled replication forks. At this stage, it is unclear if this is indeed the case. Nonetheless, combined with the technological advancements, I think this work will be of interest to the DNA-replication field.

We thank Referee 3 for his/her enthusiastic support of our work!